# Are VLMs Seeing or Just Saying?
# Uncovering the Illusion of Visual Re-examination

**Chufan Shi** [*1]  **Cheng Yang** [*2]  **Yaokang Wu** [3]
**Linghao Jin** [1]  **Bo Shui** [4]  **Taylor Berg-Kirkpatrick** [2]  **Xuezhe Ma** [1]

## Abstract

Vision-Language Models (VLMs) often produce self-reflective statements like "let me check the figure again" during reasoning. Do such statements trigger genuine visual re-examination, or are they merely learned textual patterns? We investigate this via VISUALSWAP, an image-swap probing framework: after a model reasons over an image, we replace it with a visually similar but semantically different one and test whether the model notices. We introduce VS-BENCH, 800 image pairs curated from MathVista, Math-Verse, MathVision, and MMMU-Pro. Experiments on Qwen3-VL, Kimi-VL, and ERNIE-VL reveal a striking failure: models overwhelmingly miss the swap, with accuracy dropping by up to 60%. Counterintuitively, thinking models are nearly 3x more vulnerable than their instructed counterparts, and scaling offers no mitigation. Multi-turn user instructions restore visual grounding, but self-generated reflective statements during continuous generation do not. Attention analysis explains why: user instructions substantially elevate attention to visual tokens, whereas self-reflection does not. Current VLMs tend to *say* rather than actually *see* when claiming to perform visual re-examination. Our code and dataset are available at the project page: https://visualswap.github.io/

## 1. Introduction

Vision-Language Models (VLMs) have demonstrated remarkable progress in multimodal information process-

ing (OpenAI, 2023; Team et al., 2023; Wang et al., 2024b). Recent developments in reasoning-enhanced VLMs further leverage extended chain-of-thought generation for test-time scaling, demonstrating significant improvements across diverse tasks (Bai et al., 2025a; DeepMind, 2025; Seed, 2025; Singh et al., 2025). Within this reasoning process, self-reflection (Deng et al., 2025; Madaan et al., 2023; Shinn et al., 2023; Wang et al., 2025) plays a pivotal role, enabling models to critique and refine their generated content.

For VLMs, effective self-reflection critically requires visual re-examination: the model should verify that generated content remains faithful to the input image and mitigate possible perceptual hallucinations. While contemporary VLMs can generate self-reflective statements during inference, such as "Wait, let me check the figure again", this raises a natural question: ***when models produce such statements, do they genuinely re-attend to the visual input, or merely reproduce learned linguistic patterns without true visual grounding?*** This question is critical for model optimization and trustworthiness. An illustrative scenario (Fig. 1) to this question is that, when the input image is swapped after reflective statements, can models detect visual discrepancies and correct their reasoning through genuine re-examination?

To this end, this work investigates whether VLMs authentically perform visual re-examination during self-reflective moments (Sec.2). We propose VISUALSWAP, a diagnostic framework that swaps visual inputs during the reasoning process to probe whether models genuinely re-attend to images. The framework employs a two-stage protocol: models generate reasoning from an initial image, then we replace the visual input while retaining the reasoning context and prompt reflection. Models capable of genuine visual re-examination should detect discrepancies between the swapped image and prior reasoning, maintaining performance comparable to direct inference on the new image.

To construct realistic evaluation instances, we curate VS-BENCH, a benchmark comprising 800 samples derived from challenging benchmarks including MathVista (Lu et al., 2024), MathVerse (Zhang et al., 2024), MathVision (Wang et al., 2024a), and MMMU-Pro (Yue et al., 2025) (Sec.3). For each sample, we strategically curate image pairs that

---

[*]Equal contribution  [1]University of Southern California [2]University of California San Diego [3]Carnegie Mellon University [4]University of Illinois Urbana-Champaign. Correspondence to: Chufan Shi <chufansh@usc.edu>.

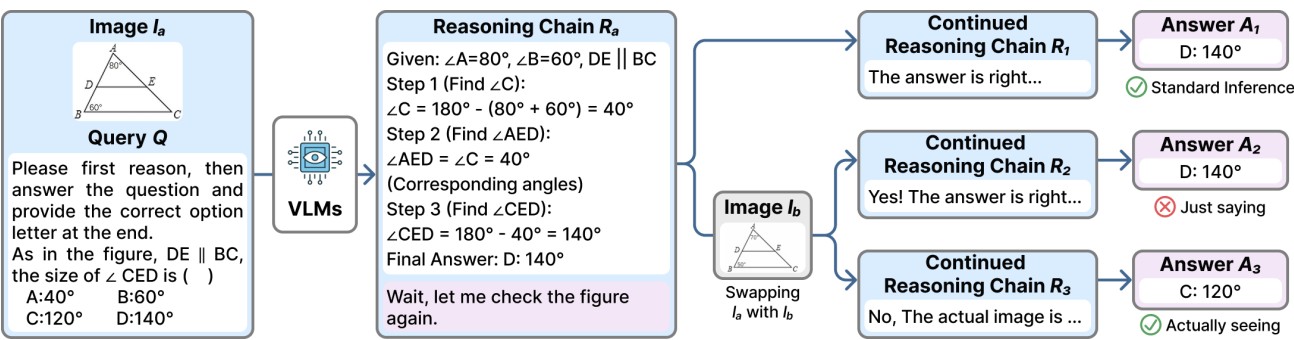

*Figure 1.* **Illustrative example of visual re-examination.** Left: Given image $I_a$ and query $Q$, the VLM generates reasoning chain $R_a$ including a self-reflective trigger. Top-right: Under standard inference with $I_a$, the model validates its logic to reach the correct answer $A_1$. Bottom-right: In the VISUALSWAP condition, $I_a$ is replaced by a visually similar but semantically distinct image $I_b$ post-reflection. Despite explicitly "saying" it will re-examine the figure, the model fails to detect the visual discrepancy and yields an incorrect answer $A_2$ grounded in the stale context of $I_a$. Only an ideal model exhibits genuine visual grounding to reach $A_3$.

share high-level visual similarity while diverging in specific fine-grained details critical for answering the question, ensuring that the two images correspond to distinct ground truth answers. This construction enables us to simulate scenarios where models must genuinely re-attend to visual content to detect discrepancies and produce correct answers.

Experiments across 15 models (Sec. 4), including Qwen3-VL, Kimi-VL, and ERNIE-VL, uncover a critical failure mode: models consistently fail to detect image changes, persisting in reasoning based on the initial visual context. This reasoning inertia leads to performance drops of up to 60% (e.g., ERNIE-4.5-VL-Thinking plummets from 79.9% to 19.6%). Counterintuitively, thinking models exhibit significantly greater vulnerability than their instruct counterparts. For instance, Qwen3-VL-32B-Thinking suffers a 48.3% drop compared to just 17.9% for the Instruct version. Furthermore, scaling offers no mitigation; larger models often fare worse (e.g., $\Delta = 54.6\%$ for Qwen3-VL-235B-A22B vs. $\Delta = 39.4\%$ for Qwen3-VL-8B).

Mechanistically, our attention analysis reveals that while self-reflective statements do elicit a marginal increase in attention to image tokens, this activation pales in comparison to user re-examination instructions (Sec. 5). In fact, the visual attention during self-reflection remains insufficient to effectively reground the reasoning process. This disparity clarifies the core issue: models are merely "saying" they will check the figure without looking closely enough to detect changes. In contrast, explicit multi-turn user instructions substantially elevate visual attention under the same context, exposing the fragility of self-initiated re-examination. Furthermore, we observe that extended reasoning contexts exacerbate the tendency to merely "declare" re-examination without execution, decoupling the model from visual evidence. While this might appear to be a simple long-context dependency issue, the continued success of multi-turn prompting, even under identical context loads, reveals a specific

limitation of the CoT mechanism: unlike external user instructions, the model's intrinsic chain-of-thought struggles to sustain visual grounding over long generations.

We highlight our contributions as follows:

- We propose VISUALSWAP, an evaluation framework to investigate whether VLMs genuinely perform visual re-examination, and curate VS-BENCH, a benchmark of 800 image pairs with visual modifications altering ground-truth answers while preserving question validity. This diagnostic approach enables direct testing of whether self-reflective statements trigger genuine visual grounding.

- Through experiments across 15 models from Qwen3-VL, Kimi-VL, and ERNIE-VL families, we uncover a severe failure mode: models fail to detect image changes during reflection, with performance degradation up to 60%. Thinking models exhibit nearly $3\times$ greater vulnerability than instruct counterparts.

- We reveal the underlying mechanism: self-reflective statements elicit insufficient visual attention, merely "saying" without "looking," whereas multi-turn instructions restore grounding. Furthermore, extended reasoning contexts decouple models from visual evidence, exposing a fundamental limitation of intrinsic CoT mechanisms.

## 2. VISUALSWAP: A Diagnostic Framework for Visual Re-examination

### 2.1. Overview

We introduce VISUALSWAP, a principled diagnostic framework designed to evaluate whether VLMs perform genuine visual re-examination during self-reflection. Our approach employs a two-stage protocol: first, models generate reasoning chains from original image-question pairs through standard inference; second, we substitute the visual input with an alternative image while retaining the generated rea-

soning context, then prompt the model to re-examine. This design directly challenges whether models anchor their responses to current visual evidence or rely predominantly on linguistic coherence from prior reasoning.

## 2.2. Problem Formulation

Each evaluation instance is defined by a triplet $\mathcal{T} = (I_a, I_b, Q)$, where $I_a$ denotes the original image and $I_b$ represents the alternative image. The image pair $(I_a, I_b)$ shares high-level visual similarity while diverging in specific fine-grained details that are critical for answering question $Q$. This divergence ensures that $I_a$ and $I_b$ correspond to distinct ground truth answers, denoted as $A_a$ and $A_b$, respectively. We use $\mathcal{M}$ to denote the VLM under evaluation.

## 2.3. Evaluation Protocol

VISUALSWAP examines whether models genuinely re-examine visual content by manipulating the consistency between visual input and reasoning context. The evaluation proceeds through two sequential stages:

**Stage 1: Standard Inference.** Given the original image $I_a$ and question $Q$, the model generates a reasoning chain $R_a$. This reasoning provides a textual description grounded in the visual content of $I_a$ and culminates in an initial answer:

$$R_a = \mathcal{M}(I_a, Q) \tag{1}$$

**Stage 2: Re-examination Probe.** We simulate the process of verifying reasoning against visual evidence by constructing a probe condition. Specifically, we append a reflection prompt $P$ (e.g., "Wait, let me check the image again") to the generated reasoning $R_a$. Crucially, we simultaneously replace the original image $I_a$ with the alternative image $I_b$. The model then continues generation conditioned on image $I_b$, question $Q$, and the combined context $R_a \oplus P$.

$$R_b = \mathcal{M}(I_b, Q, R_a \oplus P) \tag{2}$$

Effective visual re-examination requires the model to detect the discrepancy between the current visual input ($I_b$) and the prior reasoning ($R_a$), leading to a revised response. Full implementation details are provided in Appx. F.

## 2.4. Evaluation Metrics

We employ three complementary metrics to quantify visual re-examination effectiveness:

**Base Accuracy (Acc_base)** measures the model's accuracy under standard inference directly on image $I_b$ with question $Q$. It reflects the model's performance under normal reasoning conditions without any additional manipulation.

**Probe Accuracy (Acc_probe)** evaluates the model's final answer accuracy under the VISUALSWAP evaluation. It di-

rectly assesses whether the model can ground its revised reasoning $R_b$ in the current visual input $I_b$ despite prior context $R_a$, and produce the correct answer $A_b$.

**Performance Degradation ($\Delta$)** quantify the extent of visual re-examination deficiency through the performance gap:

$$\Delta = \text{Acc}_{\text{base}} - \text{Acc}_{\text{probe}} \tag{3}$$

This metric indicates insufficient visual re-examination during reflective statements. A substantial degradation ($\Delta \gg 0$) reveals that the model's revision process fails to adequately incorporate current visual evidence. Instead, the model remains predominantly anchored to linguistic patterns from prior reasoning, unable to correct its response despite having access to the appropriate visual information.

# 3. VS-BENCH: A Benchmark for Visual Re-examination

To support the comparative evaluation framework described in Sec. 2, we introduce VS-BENCH, a benchmark comprising 800 carefully constructed image pairs and corresponding questions and answers. Unlike existing benchmarks that assess static visual understanding, VS-BENCH specifically tests whether models can genuinely re-examine visual input to detect and correct perceptual inconsistency during self-reflective reasoning, a capability fundamentally distinct from conventional visual question answering.

## 3.1. Design Principles

VS-BENCH is constructed following three core principles that directly support the evaluation methodology:

**Question Invariance.** For image pair $(I_a, I_b)$, the question $Q$ must remain semantically valid and naturally applicable to both images. This ensures that performance differences in probe inference stem from visual re-examination capabilities rather than question-image compatibility issues.

**Visual Similarity.** Images $I_a$ and $I_b$ must share strong visual similarity, including layout, style, composition, and overall context. The visual similarity ensures that the reasoning chain $R_a$ generated from $I_a$ appears superficially plausible when paired with $I_b$, thereby challenging the model to conduct through visual re-examination.

**Visual Divergence.** While maintaining overall similarity, images $I_a$ and $I_b$ must contain distinct visual details that lead to different ground truth answers $A_a$ and $A_b$. This divergence enables unambiguous evaluation: successful visual re-examination should lead the model to detect the specific differences in $I_b$, resulting in a revised answer $A_b$.

*Table 1.* Image similarity metrics for pairs $(I_a, I_b)$ in VS-BENCH. Higher CLIP/SSIM and lower LPIPS indicate greater similarity.

| Source | CLIP↑ | SSIM↑ | LPIPS↓ |
|---|---|---|---|
| MathVista | 0.95 | 0.88 | 0.12 |
| MathVerse | 0.97 | 0.94 | 0.06 |
| MathVision | 0.95 | 0.86 | 0.17 |
| MMMU-Pro | 0.94 | 0.75 | 0.19 |
| **Average** | **0.95** | **0.86** | **0.14** |

### 3.2. Data Sources

We first source instances $(I_b, Q)$ from four challenging benchmarks that emphasize visual reasoning with precise, visually-determined answers:

**MathVista** (Lu et al., 2024) evaluates comprehensive multimodal mathematical reasoning by requiring models to integrate visual perception with textual logic. It includes diverse problem types such as function plots and statistical charts.

**MathVerse** (Zhang et al., 2024) tests whether models truly rely on visual input rather than textual shortcuts by minimizing redundant text descriptions. We specifically select samples from its Vision-Dominant subset, comprising plane and solid geometry problems where diagrams are essential.

**MathVision** (Wang et al., 2024a) addresses the limitations in difficulty and subject breadth of prior benchmarks by evaluating reasoning on challenging real-world math competitions. It spans 16 distinct disciplines, introducing advanced fields like topology and graph theory previously underexplored in visual contexts.

**MMMU-Pro** (Yue et al., 2025) evaluates expert-level multimodal understanding across professional disciplines, rigorously filtering out questions solvable by text alone to ensure genuine visual reasoning. We select challenging samples from fields that demand both high-level domain knowledge and precise visual interpretation.

### 3.3. Data Annotation

We employ a human-in-the-loop annotation pipeline assisted by Nano Banana Pro (Google, 2025) to generate $I_a$ based $(I_b, Q)$. For each candidate question $Q$ with image $I_b$, annotators first identify answer-critical visual elements and specify targeted modifications that will yield a different ground truth answer $A_a$ while maintaining question validity. Then, annotators leverage Nano Banana Pro to generate candidate images $I_a$, which annotators iteratively refine through feedback until quality standards are met.

To ensure controlled comparison and prevent confounding factors, we enforce that $I_a$ and $I_b$ maintain identical resolutions. This is critical because resolution differences would alter the number of visual tokens during encoding, poten-

tially introducing spurious performance variations unrelated to visual re-examination capabilities.

Through this annotation and verification process, we select 200 high-quality image pairs from each of the four data sources, yielding 800 image pairs in total. Illustrative data examples are provided in Appx. A.

### 3.4. Quality Verification

To validate that image pairs $(I_a, I_b)$ satisfy visual similarity while differing in answer-critical details, we compute three complementary metrics between $I_a$ and $I_b$: CLIP Similarity (Radford et al., 2021) for semantic alignment, SSIM (Wang et al., 2004) for structural consistency, and LPIPS (Zhang et al., 2018) for perceptual distance.

Tab. 1 reports comprehensive results across VS-BENCH. High CLIP (0.95) and SSIM (0.86), combined with low LPIPS (0.14), confirm that image pairs are visually similar yet semantically distinct—superficially confusable but with unambiguous answer differences that ensure robustness. A human study (Appx. G) further confirms the differences are reliably detectable by human observers.

## 4. Experiments

### 4.1. Experimental Setup

**Models.** We evaluate recent VLMs spanning diverse model families and scales. From the Qwen3-VL family (Bai et al., 2025a), we include Qwen3-VL-8B, Qwen3-VL-32B, Qwen3-VL-30B-A3B, and Qwen3-VL-235B-A22B. From Qwen2.5-VL (Bai et al., 2025b), we evaluate Qwen2.5-VL-7B-Instruct alongside two thinking variants: OpenVLThinker-7B (Deng et al., 2025) and VL-Rethinker-7B (Wang et al., 2025). We also include Kimi-VL-A3B (Team et al., 2025) and ERNIE-4.5-VL-28B-A3B (ERNIE, 2025), each with both Instruct and Thinking variants. All model links are provided in Appx.B. We do not evaluate closed-source models as they do not support the mechanism required for our probing experiments.

**Implementation.** For base accuracy evaluation ($\text{Acc}_{\text{Base}}$), we perform standard single-turn inference with temperature 0.1. For probe evaluation ($\text{Acc}_{\text{Probe}}$), we generate the initial reasoning $R_a$ from image $I_a$, concatenate it with the reflection prompt $P$, and continue generation with the replaced image $I_b$. We use official chat templates and default generation configurations for each model. All experiments are conducted on NVIDIA H200 GPUs. Comprehensive implementation details are provided in Appx. C.

### 4.2. Main Results

Tab. 2 presents results across all models on VISUALSWAP. For each subtask, we report base accuracy ($\text{Acc}_{\text{Base}}$), probe

accuracy ($\text{Acc}_{\text{Probe}}$), and the degradation ($\Delta$).

**All models fail to detect image changes.** Across all models and benchmarks, probe accuracy drops substantially from baseline. While models achieve high performance on standard inference, accuracy plummets when the image is swapped during self-reflection. Qwen3-VL-235B-Thinking suffers a catastrophic drop from 88.8% to 34.1%. This failure is consistent: models are explicitly prompted to "check the figure again," yet continue reasoning as if the original image were still present. The reflective prefix does not trigger genuine visual re-grounding; instead, models remain anchored to the prior reasoning trajectory, producing answers consistent with $I_a$ rather than the current image $I_b$.

**Thinking models fail more severely.** Thinking variants exhibit substantially larger degradation than instruction-tuned counterparts across all model families. For instance, Qwen3-VL-32B-Instruct drops 17.9% on average, while its Thinking variant drops 48.3%, nearly 3× the degradation. ERNIE-4.5-VL shows a similar pattern, with a drop of 34.3% for Instruct versus 60.3% for Thinking. The finding is counterintuitive: Thinking models are designed for careful reasoning, yet this capability exacerbates visual re-examination failure. Extended reasoning chains create stronger textual inertia that subsequent generation cannot overcome.

**Scale does not help.** Larger models do not mitigate the failure. Comparing Qwen3-VL-8B-Thinking ($\Delta = 39.4\%$) to Qwen3-VL-235B-A22B-Thinking ($\Delta = 54.6\%$), we observe that the 235B model exhibits greater degradation despite having orders of magnitude more parameters. Similarly, Qwen3-VL-30B-A3B-Thinking shows a severe average drop of 54.5%. Increased capacity improves baseline accuracy but does not address the fundamental limitation in visual re-examination during continuous decoding.

## 5. Discussion

### 5.1. Models Can Solve Both Images Independently

A potential confound is that alternative images $I_b$ might be inherently harder than originals $I_a$, and the observed degradation reflects task difficulty rather than re-examination failure. To rule this out, we evaluate all models on both $I_a$ and $I_b$ under standard single-turn inference without any injected context. Tab. 3 shows that models achieve comparable accuracy on $I_a$ and $I_b$ when evaluated independently. The gap between Base($I_a$) and Base($I_b$) ranges from 0.1% to 6.3% , negligible compared to the 20% to 55% degradation observed in the probe setting. This confirms that alternative images are well within model capability. The failure arises specifically when models must re-examine visual input in the presence of conflicting textual context. The same model that correctly answers a question about $I_b$ in isolation fails catastrophically when asked to verify its

reasoning after the image has been swapped. The visual understanding capability exists; what fails is the ability to deploy this capability during continuous generation.

### 5.2. Explicit User Instruction Restores Visual Grounding

Our probe experiments demonstrate that self-generated reflective prefixes fail to trigger genuine visual re-grounding. This raises a critical question: is the model fundamentally incapable of processing the new image due to context length, or is it merely suppressing visual intake due to reasoning inertia? To distinguish between capability and behavior, we investigate the efficacy of explicit multi-turn intervention.

We design a Multi-turn variant to test this. Unlike the probe setting where reflection is part of continuous generation, we structure the interaction as a two-turn conversation.

**Turn 1: Initial reasoning.** The model receives image $I_a$ and question $Q$, generating the initial reasoning chain $R_a$:

$$R_a = \mathcal{M}(I_a, Q) \tag{4}$$

**Turn 2: User-initiated re-examination.** The image is replaced with $I_b$, and the user provides a distinct instruction $U$ requesting re-examination:

$$R_{\text{multi}} = \mathcal{M}(I_b, [Q, R_a, U]) \tag{5}$$

where $[\cdot]$ denotes the multi-turn conversation history. Crucially, $U$ is an explicit user instruction (e.g., "Check the image again and re-examine"), unlike the self-generated reflection statements in the probe setting (Sec. 2).

We focus our analysis on Qwen3-VL-8B and Qwen3-VL-235B-A22B, serving as representative models for this and all subsequent analyses. Tab. 4 presents the results. Strikingly, explicit user instruction almost entirely eliminates the blindness observed in the probe setting. The recovery is particularly dramatic for the largest model: Qwen3-VL-235B-A22B-Thinking rebounds from 34.1% to 85.4% (+51.3%), nearly matching its original baseline accuracy of 88.8%.

This recovery confirms a pivotal finding: the model retains a strong latent potential to recognize visual changes, even after extended reasoning. While models fail to instruct themselves to look again within a single pass, explicit external instructions successfully unlock this capability, demonstrating that the visual information is accessible but requires human intervention to be effectively utilized. We further decompose which factor of the user instruction drives this recovery in Appx. H.

### 5.3. Attention Patterns Explain the Gap

To investigate the mechanism underlying the performance gap between self-reflection (Probe) and external intervention

*Table 2.* Main results on VS-BENCH. We report Base and Probe accuracies alongside the performance drop ($\Delta$) across 15 VLMs.

| Model | Variant | MathVista | | | MathVerse | | | MathVision | | | MMMU-Pro | | | Avg. | | |
| | | $Acc_{Base}$ | $Acc_{Probe}$ | $\Delta$ | $Acc_{Base}$ | $Acc_{Probe}$ | $\Delta$ | $Acc_{Base}$ | $Acc_{Probe}$ | $\Delta$ | $Acc_{Base}$ | $Acc_{Probe}$ | $\Delta$ | $Acc_{Base}$ | $Acc_{Probe}$ | $\Delta$ |
|---|---|---|---|---|---|---|---|---|---|---|---|---|---|---|---|---|
| Qwen3-VL-8B | Instruct | 82.5 | 55.0 | 27.5 | 70.5 | 44.0 | 26.5 | 49.5 | 31.0 | 18.5 | 74.0 | 56.5 | 17.5 | 69.1 | 46.6 | 22.5 |
| | Thinking | 84.5 | 36.5 | 48.0 | 83.0 | 29.5 | 53.5 | 56.0 | 27.0 | 29.0 | 80.5 | 53.5 | 27.0 | 76.0 | 36.6 | 39.4 |
| Qwen3-VL-32B | Instruct | 87.5 | 59.0 | 28.5 | 84.0 | 70.5 | 13.5 | 60.0 | 43.5 | 16.5 | 87.0 | 74.0 | 13.0 | 79.6 | 61.8 | 17.9 |
| | Thinking | 94.5 | 33.0 | 61.5 | 89.5 | 24.0 | 65.5 | 67.0 | 32.0 | 35.0 | 88.5 | 57.5 | 31.0 | 84.9 | 36.6 | 48.3 |
| Qwen3-VL-30B-A3B | Instruct | 84.5 | 49.5 | 35.0 | 70.5 | 55.5 | 15.0 | 49.0 | 29.5 | 19.5 | 78.0 | 64.0 | 14.0 | 70.5 | 49.6 | 20.9 |
| | Thinking | 88.5 | 18.0 | 70.5 | 89.5 | 10.0 | 79.5 | 62.5 | 24.5 | 38.0 | 84.0 | 54.0 | 30.0 | 81.1 | 26.6 | 54.5 |
| Qwen3-VL-235B-A22B | Instruct | 89.0 | 62.5 | 26.5 | 83.5 | 63.5 | 20.0 | 62.5 | 40.5 | 22.0 | 89.5 | 78.5 | 11.0 | 81.1 | 61.3 | 19.9 |
| | Thinking | 93.5 | 29.5 | 64.0 | 96.5 | 22.5 | 74.0 | 74.0 | 31.0 | 43.0 | 91.0 | 53.5 | 37.5 | 88.8 | 34.1 | 54.6 |
| ERNIE-4.5-VL-28B-A3B | Instruct | 76.0 | 33.0 | 43.0 | 71.0 | 31.5 | 39.5 | 33.5 | 18.5 | 15.0 | 72.5 | 33.0 | 39.5 | 63.3 | 29.0 | 34.3 |
| | Thinking | 87.5 | 16.5 | 71.0 | 91.0 | 13.0 | 78.0 | 60.5 | 27.5 | 33.0 | 80.5 | 21.5 | 59.0 | 79.9 | 19.6 | 60.3 |
| Kimi-VL-A3B | Instruct | 75.0 | 31.0 | 44.0 | 43.5 | 16.0 | 27.5 | 26.0 | 17.0 | 9.0 | 49.0 | 21.5 | 27.5 | 48.4 | 21.4 | 27.0 |
| | Thinking | 87.5 | 28.5 | 59.0 | 69.5 | 19.5 | 50.0 | 52.5 | 26.0 | 26.5 | 69.5 | 35.5 | 34.0 | 69.8 | 27.4 | 42.4 |
| Qwen2.5-VL-7B | Instruct | 72.5 | 37.5 | 35.0 | 49.5 | 36.0 | 13.5 | 25.5 | 15.5 | 10.0 | 47.0 | 28.5 | 18.5 | 48.6 | 29.4 | 19.3 |
| OpenVLThinker-7B | Thinking | 77.5 | 42.5 | 35.0 | 51.0 | 35.0 | 16.0 | 25.0 | 15.5 | 9.5 | 48.0 | 21.5 | 26.5 | 50.4 | 28.6 | 21.8 |
| VL-Rethinker-7B | Thinking | 79.0 | 33.0 | 46.0 | 66.0 | 32.0 | 34.0 | 31.5 | 26.5 | 5.0 | 50.5 | 18.5 | 32.0 | 56.8 | 27.5 | 29.3 |

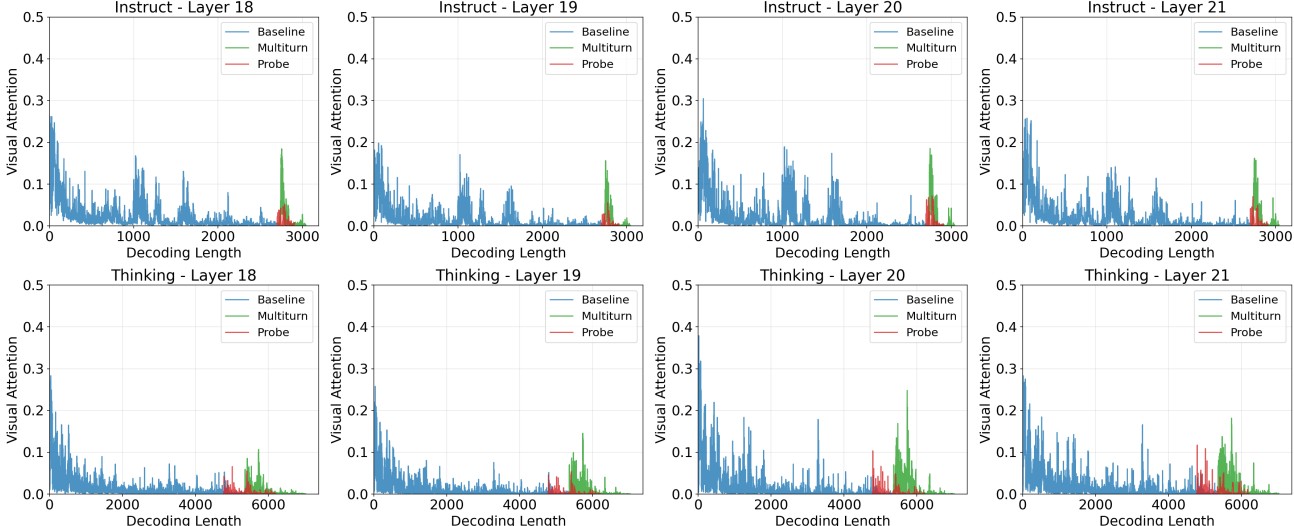

*Figure 2.* Visual attention score $S_{\text{vis}}^{(l)}$ across decoding steps for Qwen3-VL-8B (layers 18-21). Probe shows lower attention than baseline throughout generation. Multi-turn elevates attention substantially after the user instruction.

(Multi-turn), we visualize the model's attention distribution during decoding. We define the Visual Attention Score $S_{\text{vis}}^{(l)}(t)$ for a given layer $l$ at decoding step $t$ as the average attention weight allocated to image tokens across all heads:

$$S_{\text{vis}}^{(l)}(t) = \frac{1}{H} \sum_{h=1}^{H} \sum_{v \in \mathcal{V}} \mathbf{A}_{t,v}^{(l,h)} \quad (6)$$

where $H$ represents the number of attention heads, $\mathcal{V}$ denotes the set of image token indices, and $\mathbf{A}_{t,v}^{(l,h)}$ is the attention probability from the current token $t$ to image token $v$ in head $h$ of layer $l$. We report the average score across middle layers (e.g., 18-21 for 8B, 49-52 for 235B) where semantic integration typically occurs (Yin et al., 2025).

To provide a comprehensive analysis, we present qualitative evolution of attention throughout the decoding process in

Fig. 2 and 3. Furthermore, we conduct a quantitative study by sampling 100 cases, calculating the average attention score for the 100 tokens immediately before and after the intervention point (Probe or Multi-turn) in Tab. 5.

The results reveal a stark contrast between settings. In the Probe setting, self-reflective statements elicit negligible visual attention shifts. As shown in Fig. 2, the attention score remains consistently low during the "checking" phase. Quantitatively, Qwen3-VL-235B-Thinking (Layer 55) shows a minimal increase of $\Delta = 1.07$, confirming that the model acts without looking despite its textual claims. Conversely, the Multi-turn setting drives a substantial surge in visual engagement. Under identical context, the same model at Layer 55 exhibits a jump of $\Delta = 2.21$, which is more than double the activation. A similar pattern holds for Qwen3-VL-8B-Thinking ($\Delta = 2.44$ vs. $4.70$).

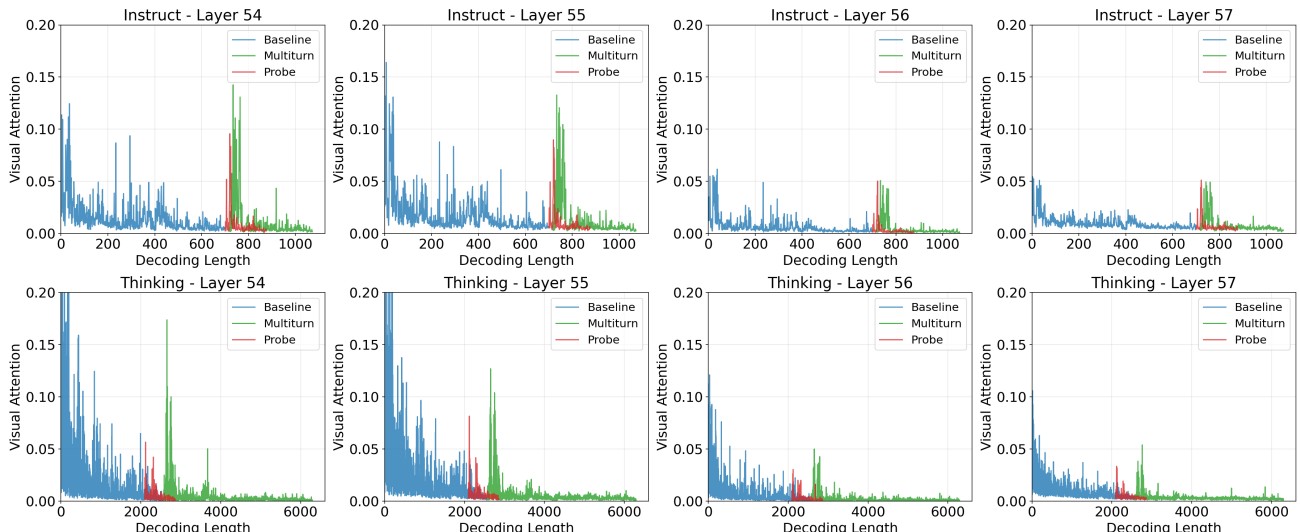

*Figure 3.* Visual attention score $S_{\text{vis}}^{(l)}$ across decoding steps for Qwen3-VL-235B-A22B (layers 54-57). The same pattern holds at scale: probe suppresses visual attention while multi-turn restores it.

*Table 3.* Average baseline accuracy on $I_a$ vs. $I_b$ under standard inference. Models perform comparably on both, confirming $I_b$ is not inherently harder. Full per-task results in Tab.11.

| Model | Variant | $I_a$ | $I_b$ | $\Delta$ |
|---|---|---|---|---|
| Qwen3-VL-8B | Instruct | 69.1 | 67.5 | 1.6 |
| | Thinking | 76.0 | 75.8 | 0.3 |
| Qwen3-VL-32B | Instruct | 79.6 | 79.5 | 0.1 |
| | Thinking | 84.9 | 81.8 | 3.1 |
| Qwen3-VL-30B-A3B | Instruct | 70.5 | 72.5 | -2.0 |
| | Thinking | 81.1 | 79.0 | 2.1 |
| Qwen3-VL-235B-A22B | Instruct | 81.1 | 82.8 | -1.6 |
| | Thinking | 88.8 | 84.4 | 4.4 |
| ERNIE-4.5-VL-28B-A3B | Instruct | 63.3 | 57.0 | 6.3 |
| | Thinking | 79.9 | 76.0 | 3.9 |
| Kimi-VL-A3B | Instruct | 48.4 | 46.3 | 2.1 |
| | Thinking | 69.8 | 65.4 | 4.4 |
| Qwen2.5-VL-7B | Instruct | 48.6 | 49.5 | -0.9 |
| OpenVLThinker-7B | Thinking | 50.4 | 47.5 | 2.9 |
| VL-Rethinker-7B | Thinking | 56.8 | 51.3 | 5.5 |

*Table 4.* Average accuracy under Probe vs. Multi-turn. Explicit user instruction substantially recovers performance. Full per-benchmark results in Tab. 12.

| Model | Variant | Base | Probe | Multi-turn |
|---|---|---|---|---|
| Qwen3-VL-8B | Instruct | 69.1 | 46.6 | 58.2 |
| | Thinking | 76.0 | 36.6 | 67.5 |
| Qwen3-VL-235B-A22B | Instruct | 81.1 | 61.3 | 77.9 |
| | Thinking | 88.8 | 34.1 | 85.4 |

penetrates this textual barrier to re-engage the attention.

### 5.4. Impact of Reasoning Context Length

To investigate the impact of reasoning length, we systematically vary the retained initial reasoning chain ($R_a$) at 0%, 25%, 50%, 75% and 100% before the image swap. Fig. 4 reveals a consistent monotonic decline in probe accuracy across all models. The degradation is particularly severe for Thinking models: Qwen3-VL-235B-Thinking plummets from 88.8% (0% context) to 34.1% (100% context), a staggering 54.7% drop. These results confirm that longer reasoning chains induce stronger contextual inertia, progressively decoupling the model from visual input. However, this suppression is reversible. As noted in Tab. 4, multi-turn interaction restores accuracy to 85.4% even with full context history. This demonstrates that while continuous decoding traps the model in its own trajectory, the underlying visual capability remains intact via explicit external intervention.

### 5.5. Robustness to Prompt Variations

To ensure that the observed failure is not an artifact of specific phrasing, we evaluate robustness across 10 semantically

**The Mechanism of Failure.** This attentional disparity provides a mechanistic explanation for our main results. The performance gap is not due to a lack of visual capability, as the model can attend to the image when forced. Instead, it stems from a failure of autonomous attentional control. This failure is particularly acute in Thinking models: as the chain-of-thought extends, the model becomes increasingly decoupled from the visual tokens, relying instead on the strong prior of its own generated text. Consequently, the "checking" action is superficial, acting as a simulation of the linguistic form of re-examination without the neural execution. Only an explicit external interrupt effectively

*Table 5.* Visual attention scores under Probe vs. Multi-turn across different layers. Explicit user instruction substantially recovers performance. We report scores averaged over 100 tokens before (Bef.) and after (Aft.) the prompt for both Probe and Multi-turn (MT) settings. Higher values indicate stronger attention to image tokens.

| Model | Variant | Layer A | | | | | | Layer B | | | | | | Layer C | | | | | | Layer D | | | | | |
|---|---|---|---|---|---|---|---|---|---|---|---|---|---|---|---|---|---|---|---|---|---|---|---|---|---|
| | | Probe | | | MT | | | Probe | | | MT | | | Probe | | | MT | | | Probe | | | MT | | |
| | | Bef. | Aft. | Δ | Bef. | Aft. | Δ | Bef. | Aft. | Δ | Bef. | Aft. | Δ | Bef. | Aft. | Δ | Bef. | Aft. | Δ | Bef. | Aft. | Δ | Bef. | Aft. | Δ |
| Qwen3-VL-8B | | *(L18)* | | | | | | *(L19)* | | | | | | *(L20)* | | | | | | *(L21)* | | | | | |
| | Instruct | 2.23 | 3.97 | 1.73 | 2.09 | 5.90 | 3.81 | 1.81 | 3.06 | 1.26 | 1.70 | 4.36 | 2.66 | 2.05 | 4.45 | 2.40 | 2.19 | 6.46 | 4.27 | 2.24 | 4.72 | 2.48 | 2.44 | 6.77 | 4.33 |
| | Thinking | 2.27 | 3.92 | 1.65 | 1.46 | 4.71 | 3.24 | 2.16 | 3.33 | 1.17 | 1.55 | 4.00 | 2.45 | 2.36 | 4.66 | 2.30 | 1.67 | 5.64 | 3.97 | 2.54 | 4.98 | 2.44 | 1.80 | 6.49 | 4.70 |
| Qwen3-VL-235B-A22B | | *(L54)* | | | | | | *(L55)* | | | | | | *(L56)* | | | | | | *(L57)* | | | | | |
| | Instruct | 1.23 | 2.24 | 1.02 | 1.13 | 3.54 | 2.41 | 1.42 | 2.64 | 1.22 | 1.34 | 3.99 | 2.65 | 0.64 | 1.22 | 0.58 | 0.62 | 1.68 | 1.07 | 0.84 | 1.33 | 0.49 | 0.88 | 1.72 | 0.84 |
| | Thinking | 1.10 | 1.96 | 0.86 | 0.50 | 2.49 | 2.00 | 1.35 | 2.41 | 1.07 | 0.65 | 2.85 | 2.21 | 0.55 | 0.99 | 0.45 | 0.30 | 1.28 | 0.98 | 0.81 | 1.20 | 0.39 | 0.50 | 1.21 | 0.71 |

*Table 6.* Probe accuracy across 10 prompt paraphrase variants. Low standard deviation confirms that performance degradation is consistent regardless of prompt phrasing.

| Model | Variant | Probe Acc. (Avg. ± Std.) |
|---|---|---|
| Qwen3-VL-8B | Instruct | 45.2 ± 3.1 |
| | Thinking | 38.9 ± 3.8 |
| Qwen3-VL-235B-A22B | Instruct | 59.1 ± 4.0 |
| | Thinking | 36.5 ± 4.9 |

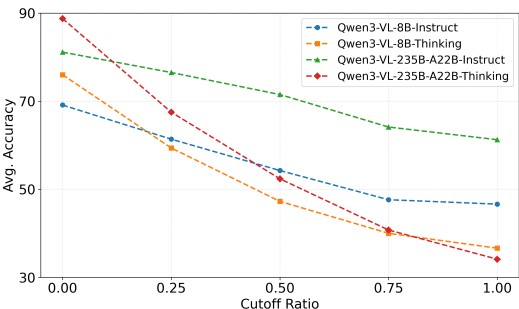

*Figure 4.* Probe accuracy vs. retained $R_a$. Performance declines monotonically as length increases, particularly for Thinking models. Per-task results are detailed in Tab. 13.

equivalent but lexically diverse reflective triggers. These variants range from casual prompts (e.g., "Wait, let me look at the image again") to formal directives (e.g., "Let me verify by examining the visual information once more"). The complete list is provided in Appx. E.

Tab. 6 demonstrates remarkable consistency across all variants. The standard deviations are negligible ($\sigma \leq 1.1$ across all models), with Qwen3-VL-235B-Thinking showing an exceptionally tight bound of $36.5 \pm 4.9$. This invariance confirms that the model's inability to update its visual grounding is a fundamental structural limitation of continuous decoding paradigm, rather than a sensitivity to specific prompts.

### 5.6. Probing Setup Reflects Natural Behavior

To verify that the observed failure reflects a genuine property of self-reflection rather than an artifact of forced trigger injection, we examine whether our probing setup faithfully aligns with how VLMs naturally engage in self-reflective reasoning. We approach this question from two complemen-

*Table 7.* Frequency of self-reflective triggers in natural generation. Reflection is prevalent across all variants, confirming the probing setup is in-distribution.

| Model | Variant | Freq. (%) |
|---|---|---|
| Qwen3-VL-8B | Instruct | 87.1 |
| | Thinking | 93.0 |
| Qwen3-VL-235B-A22B | Instruct | 90.9 |
| | Thinking | 96.9 |

tary angles: the prevalence of reflective triggers in natural generation, and the consistency of failure when swaps are performed at naturally produced reflection points.

**Reflective triggers arise naturally.** We first quantify how often models spontaneously produce reflective statements during reasoning. For Thinking models, we directly count the occurrence of reflective triggers (e.g., "wait") in their natural reasoning chains. For Instruct models, we elicit reflection via a system prompt and verify that this changes baseline accuracy by less than 2%. As shown in Tab. 7, reflective triggers appear in 87%–97% of generations across all variants, confirming that our probing operates on linguistic patterns the models routinely produce themselves.

**Swapping at natural triggers yields stronger failure.** We further conduct a stricter test: for generations that naturally contain a reflective trigger, we swap the image at that exact natural reflection point rather than appending a standardized prompt. For the remaining cases, we fall back to standard probing. Tab. 8 shows that swapping at natural reflection points yields accuracy at or below standard probing across all variants, indicating that the observed failure is not an artifact of distributional mismatch introduced by appended prompts. We adopt the standardized injection in our main experiments solely to enable fair cross-model comparison, as the lexical forms of natural reflection are too diverse to exhaustively enumerate.

### 5.7. Amplifying Visual Attention Mitigates the Illusion

Our attention analysis (Sec. 5.3) identifies insufficient visual attention as the mechanistic cause of re-examination failure. This naturally suggests a direct intervention: if the model fails to autonomously elevate attention to image tokens dur-

*Table 8.* Probe accuracy under standard injection vs. swapping at naturally generated reflection points. Natural triggers yield equal or lower accuracy, indicating that the failure persists under natural reasoning dynamics.

| Model | Variant | Standard | Natural |
|---|---|---|---|
| Qwen3-VL-8B | Instruct | 46.6 | 34.9 |
| | Thinking | 36.6 | 35.2 |
| Qwen3-VL-235B-A22B | Instruct | 61.3 | 53.1 |
| | Thinking | 34.1 | 35.3 |

*Table 9.* Probe accuracy under attention amplification ($2\times$ on image tokens). Amplifying visual attention at the reflection point recovers performance, especially for the Thinking variant.

| Variant | Probe | + Amplification | $\Delta$ |
|---|---|---|---|
| Qwen3-VL-8B-Instruct | 46.6 | 54.5 | +7.9 |
| Qwen3-VL-8B-Thinking | 36.6 | 54.8 | +18.2 |

ing self-reflection, can we elevate it externally? We test this hypothesis with a minimal training-free intervention.

**Setup.** Under the Probe setting, we scale the attention weights allocated to image tokens by a factor of $2\times$ during the generation of $R_b$, applied uniformly across all layers and heads. All other components remain unchanged.

**Findings.** Tab. 9 reveals that amplifying visual attention yields a substantial accuracy gain, with the Thinking variant improving by $+18.2\%$. This provides direct causal evidence that insufficient visual attention, rather than capability loss, drives the failure mode: forcing the model to "see" partially restores grounding without any parameter updates or additional training. The disproportionate gain on the Thinking variant further corroborates our finding that thinking models suffer from stronger attentional decoupling under extended reasoning. We view this as a proof-of-concept and leave further exploration to future work, such as incorporating multi-turn visual verification data into SFT/RL pipelines, where models are explicitly trained to detect discrepancies between prior reasoning and updated visual input, or using visual attention strength as an auxiliary reward signal to encourage genuine re-grounding during reflection.

## 6. Related Work

**Vision-Language Models.** Conventional VLMs employ distinct unimodal encoders with cross-modal fusion modules (Li et al., 2021; Lu et al., 2019; Tan & Bansal, 2019), achieving strong performance on specific tasks such as visual question answering (Goyal et al., 2017; Hudson & Manning, 2019) and image retrieval (Lin et al., 2014; Plummer et al., 2015). However, their ability to generalize to open-domain generation and complex reasoning remains constrained. Recent advances in LLMs (Bai et al., 2022; Ouyang et al., 2022; Touvron et al., 2023) have demon-

strated remarkable capabilities in instruction following and generalization, motivating a paradigm shift in VLM design. Modern VLMs leverage LLMs as the textual backbone, employing modality alignment modules to project visual features into the LLM's semantic space (Alayrac et al., 2022; Li et al., 2023; Liu et al., 2023). This approach enables VLMs to inherit the strong capabilities of LLMs while supporting versatile multimodal understanding and generation (OpenAI, 2023; Team et al., 2023; Wang et al., 2024b).

**Reasoning in VLMs.** Building upon the success of reasoning techniques in enhancing LLM performance on complex tasks (Guo et al., 2025; OpenAI, 2024; Wei et al., 2022), recent work has sought to equip VLMs with similar reasoning capabilities. Current approaches span from constructing large-scale visual instruction tuning datasets that incorporate reasoning chains (Chen et al., 2024; Liu et al., 2024a;b) to applying reinforcement learning to align chain-of-thought reasoning (Huang et al., 2025; Liu et al., 2025; Luo et al., 2025; 2026; Shen et al., 2025; Yang et al., 2025b). These methods enable contemporary VLMs to exhibit long-horizon reasoning behaviors in their textual outputs, including self-reflection (Deng et al., 2025; Wang et al., 2025; Yang et al., 2025a). However, a critical question remains unexplored: when VLMs display self-reflective behaviors, are they genuinely re-examining visual content? To address this question, we introduce the VISUALSWAP, which replaces visual inputs to probe whether models truly ground reflections in visual perception.

## 7. Conclusion

We investigate the authenticity of visual re-examinatioin in VLMs using our VISUALSWAP framework. We uncover a critical "illusion of visual re-examination": while models frequently claim to check images again, they remain blinded by textual reasoning inertia during continuous decoding. This effect is most severe in thinking models, with Qwen3-VL-235B-Thinking suffering a catastrophic accuracy drop from 88.8% to 34.1%. Crucially, our analysis reveals that this is a failure of control, not capability. While self-reflection fails to mobilize visual attention, explicit user intervention successfully breaks the textual inertia and restores visual grounding to near-baseline levels. We conclude that current CoT paradigms prioritize textual coherence over visual intake, highlighting the urgent need for future mechanisms that enable genuine autonomous attentional control.

## Acknowledgement

This work is funded in part by the Schmidt Foundation and by the National Science Foundation under grant 2146151.

## Impact Statement

This paper aims to advance the reliability and transparency of Vision-Language Models. Our findings regarding the "illusion of visual re-examination" highlight a potential risk in safety-critical domains (e.g., medical diagnostics, autonomous navigation), where users may place unwarranted trust in a model's stated verification process despite its failure to actually process visual data. By identifying the mechanism of reasoning inertia, our work seeks to mitigate these deceptive hallucinations and promote the development of AI systems with genuine attentional control. We do not foresee negative societal consequences from this diagnostic research.

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

# A. Case Study

To provide concrete illustrations of the mechanisms underlying visual re-examination—and its frequent failure—we present a qualitative analysis of representative examples from VISUALSWAP. We categorize these examples across four distinct domains: Geometry, Chart Understanding, Synthetic Scene VQA, and Function Plots.

## A.1. The Illusion of Re-examination

Fig. 5 through 8 illustrate the dominant failure mode: *Textual Inertia*. In these instances, the self-reflective trigger ("Let me check the figure again") fails to interrupt the probability distribution established by the initial reasoning chain $R_a$.

- **Geometry (Fig. 5):** The model initially calculates an angle based on a $60°$ input. When the image is swapped to $50°$, the model ignores the clear visual evidence. It hallucinates the original numerical value to maintain consistency with its prior calculation ($180° - 60°$), resulting in a derivation that contradicts the pixel-level information.

- **Chart Understanding (Fig. 6):** In the bar chart task, the model fails to notice that the "Burlywood" bar has been visually shortened to become the minimum. Instead, it relies on the specific measurements extracted during $R_a$, hallucinating the previous bar lengths to justify an incorrect "No" response.

- **Synthetic Scene (Fig. 7):** For the counting task, the prompt requires subtracting purple objects. When the purple cube is swapped for a cyan one in $I_b$, the model fails to update its object inventory. It continues to count the now-absent purple cube, demonstrating that the reasoning is anchored in the memory of $I_a$ rather than the current observation of $I_b$.

- **Function Plots (Fig. 8):** The model ignores a structural change in performance curves where "Dense" is outperformed by "Soft". Despite the visual crossover of lines in $I_b$, the model restates the textual conclusion from $R_a$, confirming that the reflection was merely linguistic.

## A.2. Evidence of Genuine Grounding

Fig. 9 through 12 demonstrate that models *possess* the capability for visual re-examination, even if it is frequently suppressed by autoregressive generation.

- **Geometry (Fig. 9):** Here, the model successfully breaks textual inertia. It explicitly identifies the new angle ($50°$) in $I_b$, distinct from the $60°$ in its history, and correctly re-derives the supplementary angle as $130°$.

- **Chart Understanding (Fig. 10):** The model correctly perceives that the "Black" bar, previously the longest, has been shortened in $I_b$. It updates its comparison logic to align with the new visual reality.

- **Synthetic Scene (Fig. 11):** In this subtraction task, the model accurately detects the removal of the red sphere (replaced by green). Crucially, it updates the arithmetic operation—subtracting zero red items instead of one—leading to the correct final count.

- **Function Plots (Fig. 12):** This case highlights a dramatic visual correction. The model transitions from analyzing a parabola ($x^2$) in $I_a$ to an absolute value function ($|2x - 3| + 1$) in $I_b$. It abandons the derivative calculation for a smooth curve and correctly identifies the constant slope properties of the V-shaped graph.

# B. Models

Table. 10 summarizes the models we use and their Hugging Face repositories.

*Table 10.* List of models and their Hugging Face repositories.

| Model Name | Hugging Face Repository |
| --- | --- |
| Qwen3-VL-8B-Instruct | https://huggingface.co/Qwen/Qwen3-VL-8B-Instruct |
| Qwen3-VL-8B-Thinking | https://huggingface.co/Qwen/Qwen3-VL-8B-Thinking |
| Qwen3-VL-32B-Instruct | https://huggingface.co/Qwen/Qwen3-VL-32B-Instruct |
| Qwen3-VL-32B-Thinking | https://huggingface.co/Qwen/Qwen3-VL-32B-Thinking |
| Qwen3-VL-30B-A3B-Instruct | https://huggingface.co/Qwen/Qwen3-VL-30B-A3B-Instruct |
| Qwen3-VL-30B-A3B-Thinking | https://huggingface.co/Qwen/Qwen3-VL-30B-A3B-Thinking |
| Qwen3-VL-235B-A22B-Instruct | https://huggingface.co/Qwen/Qwen3-VL-235B-A22B-Instruct |
| Qwen3-VL-235B-A22B-Thinking | https://huggingface.co/Qwen/Qwen3-VL-235B-A22B-Thinking |
| ERNIE-4.5-VL-28B-A3B-Instruct | https://huggingface.co/baidu/ERNIE-4.5-VL-28B-A3B-Base-PT |
| ERNIE-4.5-VL-28B-A3B-Thinking | https://huggingface.co/baidu/ERNIE-4.5-VL-28B-A3B-Thinking |
| Kimi-VL-A3B-Instruct | https://huggingface.co/moonshotai/Kimi-VL-A3B-Instruct |
| Kimi-VL-A3B-Thinking | https://huggingface.co/moonshotai/Kimi-VL-A3B-Thinking-2506 |
| Qwen2.5-VL-7B | https://huggingface.co/Qwen/Qwen2.5-VL-7B-Instruct |
| OpenVLThinker-7B | https://huggingface.co/ydeng9/OpenVLThinker-7B |
| VL-Rethinker-7B | https://huggingface.co/TIGER-Lab/VL-Rethinker-7B |

# C. Experimental Setup

**Inference Configuration.** All model inferences are conducted using the vLLM (Duan et al., 2024) on NVIDIA H200 GPUs. We employ a sampling temperature of $\tau = 0.1$. This value was empirically selected to balance reproducibility and generation quality: higher temperatures introduce excessive stochasticity that confounds the measurement of visual re-examination, while lower temperatures (e.g., greedy decoding) frequently lead to repetition loops and degeneration in thinking models (Shi et al., 2024). To ensure result stability, we performed 10 independent sampling runs on the Qwen3-VL series, observing a standard deviation in accuracy of less than 0.02, confirming that our findings are statistically robust.

**Evaluation Protocol.** We utilize VLMEvalKit (Duan et al., 2024) for standardized evaluation and answer extraction. For the LLM-as-a-Judge component, we employ Qwen3-VL-235B-A22B-Instruct instead of the conventional GPT-4o-mini.

**Prompts.** For the Probe Setting (Self-Reflection), we append the statement "Wait, let me check the figure again to make sure I haven't made a mistake." to the model's generated reasoning. For the Multi-turn Setting (User Intervention), we provide the explicit user instruction "Check the image again and re-examine." in the second turn.

# D. Per-Task Baseline Results

## D.1. Detailed Baseline Results

To validate that the performance degradation observed in our probing experiments stems from re-examination failure rather than the inherent difficulty of the counterfactual images, we evaluated all models on both the original images ($I_a$) and the swapped images ($I_b$) independently.

Tab. 11 presents the comprehensive per-benchmark breakdown. Across all datasets (MathVista, MathVerse, MathVision, and MMMU-Pro), the performance gap $\Delta$ between $I_a$ and $I_b$ remains negligible. For instance, Qwen3-VL-235B-A22B-Thinking achieves 93.5% on MathVista ($I_a$) and 91.0% on ($I_b$), a variance well within normal fluctuation. This confirms that the counterfactual images are well within the models' capabilities, reinforcing our conclusion that the massive drops in the Probe setting are due to *Reasoning Inertia*.

## D.2. Detailed Multi-turn Results

We demonstrated that explicit user instructions can restore visual grounding. Tab. 12 provides the detailed performance comparison between the Standard Baseline, the Probe setting (self-reflection), and the Multi-turn setting (user intervention) across all four benchmarks.

The results show a consistent pattern: while the Probe setting leads to catastrophic failure (e.g., Qwen3-VL-235B-Thinking

*Table 11.* Per-benchmark baseline accuracy on $I_a$ vs. $I_b$. Models perform comparably on both images across all benchmarks.

| Model | Variant | MathVista | | | MathVerse | | | MathVision | | | MMMU-Pro | | |
|---|---|---|---|---|---|---|---|---|---|---|---|---|---|
| | | $I_a$ | $I_b$ | $\Delta$ | $I_a$ | $I_b$ | $\Delta$ | $I_a$ | $I_b$ | $\Delta$ | $I_a$ | $I_b$ | $\Delta$ |
| Qwen3-VL-8B | Instruct | 82.5 | 78.0 | 4.5 | 70.5 | 71.0 | -0.5 | 49.5 | 45.5 | 4.0 | 74.0 | 75.5 | -1.5 |
| | Thinking | 84.5 | 84.0 | 0.5 | 83.0 | 86.5 | -3.5 | 56.0 | 49.0 | 7.0 | 80.5 | 83.5 | -3.0 |
| Qwen3-VL-32B | Instruct | 87.5 | 86.5 | 1.0 | 84.0 | 89.5 | -5.5 | 60.0 | 55.0 | 5.0 | 87.0 | 87.0 | 0.0 |
| | Thinking | 94.5 | 87.5 | 7.0 | 89.5 | 91.5 | -2.0 | 67.0 | 59.0 | 8.0 | 88.5 | 89.0 | -0.5 |
| Qwen3-VL-30B-A3B | Instruct | 84.5 | 86.0 | -1.5 | 70.5 | 73.0 | -2.5 | 49.0 | 52.0 | -3.0 | 78.0 | 79.0 | -1.0 |
| | Thinking | 88.5 | 88.5 | 0.0 | 89.5 | 92.0 | -2.5 | 62.5 | 52.5 | 10.0 | 84.0 | 83.0 | 1.0 |
| Qwen3-VL-235B-A22B | Instruct | 89.0 | 90.5 | -1.5 | 83.5 | 87.0 | -3.5 | 62.5 | 61.5 | 1.0 | 89.5 | 92.0 | -2.5 |
| | Thinking | 93.5 | 91.0 | 2.5 | 96.5 | 94.5 | 2.0 | 74.0 | 62.0 | 12.0 | 91.0 | 90.0 | 1.0 |
| ERNIE-4.5-VL-28B-A3B | Instruct | 76.0 | 71.5 | 4.5 | 71.0 | 70.5 | 0.5 | 33.5 | 21.5 | 12.0 | 72.5 | 64.5 | 8.0 |
| | Thinking | 87.5 | 83.5 | 4.0 | 91.0 | 90.0 | 1.0 | 60.5 | 48.5 | 12.0 | 80.5 | 82.0 | -1.5 |
| Kimi-VL-A3B | Instruct | 75.0 | 67.5 | 7.5 | 43.5 | 46.5 | -3.0 | 26.0 | 25.5 | 0.5 | 49.0 | 45.5 | 3.5 |
| | Thinking | 87.5 | 77.0 | 10.5 | 69.5 | 71.5 | -2.0 | 52.5 | 41.0 | 11.5 | 69.5 | 72.0 | -2.5 |
| Qwen2.5-VL-7B | Instruct | 72.5 | 67.0 | 5.5 | 49.5 | 58.5 | -9.0 | 25.5 | 27.0 | -1.5 | 47.0 | 45.5 | 1.5 |
| OpenVLThinker-7B | Thinking | 77.5 | 66.5 | 11.0 | 51.0 | 52.0 | -1.0 | 25.0 | 25.0 | 0.0 | 48.0 | 46.5 | 1.5 |
| VL-Rethinker-7B | Thinking | 79.0 | 74.5 | 4.5 | 66.0 | 59.5 | 6.5 | 31.5 | 26.5 | 5.0 | 50.5 | 44.5 | 6.0 |

*Table 12.* Per-benchmark accuracy under $\text{Acc}_{\text{Base}}$, Probe, and Multi-turn settings.

| Model | Variant | MathVista | | | MathVerse | | | MathVision | | | MMMU-Pro | | |
|---|---|---|---|---|---|---|---|---|---|---|---|---|---|
| | | Base | Probe | Multi | Base | Probe | Multi | Base | Probe | Multi | Base | Probe | Multi |
| Qwen3-VL-8B | Instruct | 82.5 | 55.0 | 68.4 | 70.5 | 44.0 | 53.5 | 49.5 | 31.0 | 42.0 | 74.0 | 56.5 | 69.0 |
| | Thinking | 84.5 | 36.5 | 71.5 | 83.0 | 29.5 | 77.0 | 56.0 | 27.0 | 47.0 | 80.5 | 53.5 | 74.5 |
| Qwen3-VL-235B-A22B | Instruct | 89.0 | 62.5 | 83.2 | 83.5 | 63.5 | 82.0 | 62.5 | 40.5 | 55.0 | 89.5 | 78.5 | 91.5 |
| | Thinking | 93.5 | 29.5 | 83.2 | 96.5 | 22.5 | 97.0 | 74.0 | 31.0 | 71.5 | 91.0 | 53.5 | 90.0 |

drops to 22.5% on MathVerse), the Multi-turn interaction consistently recovers performance to near-baseline levels (rebounding to 97.0% on the same task). This recovery is universal across benchmarks, confirming that the "illusion of visual re-examination" is a systemic issue in continuous decoding, while the underlying visual capability remains intact and accessible via external prompting.

### D.3. Detailed Context Length Ablation

To further investigate the relationship between reasoning chain length and visual decoupling, we systematically varied the retained context ratio from 0% to 100%. Tab. 13 details the Probe accuracy for Qwen3-VL-8B and Qwen3-VL-235B-A22B across each benchmark individually.

The data reveals a strictly monotonic decline in almost every case. For example, on MathVision, Qwen3-VL-235B-Thinking's accuracy decays from 74.0% (0% context) to 31.0% (100% context). This trend validates our hypothesis that longer reasoning chains induce stronger contextual inertia, progressively isolating the model from the visual encoder. The degradation is consistently more severe for Thinking models compared to their Instruct counterparts across all tasks.

## E. Reflection Prompt Variants

Tab. 14 lists all 10 reflection prompt variants used in the robustness analysis. These prompts are semantically equivalent but lexically distinct, designed to test whether the visual re-examination failure is sensitive to specific prompt phrasing.

## F. Implementation Protocol for Image Swapping

To clarify the technical realization of our image-swap mechanism, we describe the exact token-level input/output sequences for the three settings used throughout the paper. Crucially, all swap operations are implemented via *full re-prefill* of the entire context with the swapped image, rather than hidden-state intervention or KV-cache manipulation. This means that when the image is replaced, the model recomputes fresh visual representations from scratch, and has full access to the new visual information when continuing generation.

*Table 13.* Per-task impact of reasoning context length on visual re-examination. We report Probe accuracy as the retained context ratio increases from 0% to 100%. Performance consistently degrades as more reasoning context is provided, with Thinking models showing the sharpest decline.

| Model | Variant | MathVista | | | | | MathVerse | | | | | MathVision | | | | | MMMU-Pro | | | | |
|---|---|---|---|---|---|---|---|---|---|---|---|---|---|---|---|---|---|---|---|---|---|
| | | 0.00 | 0.25 | 0.50 | 0.75 | 1.0 | 0.00 | 0.25 | 0.50 | 0.75 | 1.0 | 0.00 | 0.25 | 0.50 | 0.75 | 1.0 | 0.00 | 0.25 | 0.50 | 0.75 | 1.0 |
| Qwen3-VL-8B | Instruct | 82.5 | 76.5 | 71.5 | 58.5 | 55.0 | 70.5 | 58.0 | 50.5 | 45.0 | 44.0 | 49.5 | 42.5 | 33.5 | 31.0 | 31.0 | 74.0 | 68.5 | 61.5 | 56.0 | 56.5 |
| | Thinking | 84.5 | 73.0 | 54.0 | 43.0 | 36.5 | 83.0 | 56.0 | 41.5 | 32.5 | 29.5 | 56.0 | 40.0 | 34.0 | 30.5 | 27.0 | 80.5 | 68.5 | 59.5 | 54.0 | 53.5 |
| Qwen3-VL-235B-A22B | Instruct | 89.0 | 84.0 | 77.0 | 68.5 | 62.5 | 83.5 | 80.5 | 77.0 | 67.5 | 63.5 | 62.5 | 56.5 | 52.0 | 45.0 | 40.5 | 89.5 | 85.0 | 80.0 | 75.5 | 78.5 |
| | Thinking | 93.5 | 75.0 | 53.5 | 40.5 | 29.5 | 96.5 | 58.5 | 39.0 | 27.0 | 22.5 | 74.0 | 52.5 | 47.0 | 34.5 | 31.0 | 91.0 | 84.0 | 70.0 | 61.0 | 53.5 |

*Table 14.* Reflection prompt variants used in the probe stage. All prompts convey the same intent of re-examining the image but differ in lexical expression.

| ID | Reflection Prompt |
|---|---|
| 1 | Actually, let me zoom in mentally on the image to verify the correctness. |
| 2 | Hold on, let me validiate my thought process by looking at the figure again. |
| 3 | Actually, let me review the fine details in the figure to be absolutely sure. |
| 4 | I'd better look at the input again to confirm that this observation is correct. |
| 5 | Let me pause and re-evaluate the entire figure to ensure my interpretation is sound. |
| 6 | Actually, let me zoom in on the fine-grained details to verify this specific part. |
| 7 | Let me scrutinize the figure one more time to confirm my initial impression. |
| 8 | Wait, let me take a second look at the image to ensure I'm not misinterpreting it. |
| 9 | I'd better double-check the visual input to avoid any potential perceptual errors. |
| 10 | Let me double-check the image to make sure I didn't imagine that detail. |

**(1) Standard Inference.** The model receives image $I_a$ and question $Q$, producing reasoning $R_a$ in a single assistant turn:

- **Input:**
  `[User_Start]` $I_a\, Q$ `[User_End]`

- **Output:**
  `[Response_Start]` $R_a$ `[Response_End]`

**(2) Re-examination Probe.** We replace $I_a$ with $I_b$ and re-prefill the entire prompt, keeping the previously generated reasoning $R_a$ followed by the reflection prompt $P$ *inside the same assistant turn* (i.e., without closing the response):

- **Input:**
  `[User_Start]` $I_b\, Q$ `[User_End][Response_Start]` $R_a\, P$

- **Output:**
  $R_b$ `[Response_End]`

Here $R_a$ and the continued generation $R_b$ share the same assistant response. The visual tokens of $I_b$ are freshly encoded during re-prefill.

**(3) Multi-turn.** We close $R_a$ as a complete assistant response and append a new user turn containing the explicit instruction $U$:

- **Input:**
  `[User_Start]` $I_b\, Q$ `[User_End][Response_Start]` $R_a$ `[Response_End][User_Start]` $U$ `[User_End]`

- **Output:**
  `[Response_Start]` $R_b$ `[Response_End]`

The only difference from the Probe setting is the placement of turn boundaries: in Probe, $R_a$ and $R_b$ are continuous generation within one assistant response; in Multi-turn, $R_a$ is closed and $R_b$ is generated in response to a new user turn. The visual input ($I_b$) and the textual context ($R_a$, the request to re-examine) are otherwise identical between the two settings, making this an isolated test of the role of turn structure.

*Table 15.* Accuracy under five interventions designed to isolate which factor of the Multi-turn setting drives the recovery. Random meaningless strings cause complete degeneration; structural markers alone yield negligible change; only the combination of turn boundaries and semantically meaningful content (Multi-turn) fully restores performance.

| Setting | Qwen3-VL-8B | | Qwen3-VL-235B-A22B | |
|---|---|---|---|---|
| | Instruct | Thinking | Instruct | Thinking |
| Probe (Natural Language) | 46.6 | 36.6 | 61.3 | 34.1 |
| Probe (High PPL, meaningful) | 48.4 | 43.3 | 63.1 | 47.8 |
| Probe (High PPL, meaningless) | 0.0 | 0.0 | 0.0 | 0.0 |
| Probe (System token only) | 48.6 | 33.3 | 64.0 | 37.9 |
| Multi-turn (Natural Language) | 58.2 | 67.5 | 77.9 | 85.4 |

## G. Human Baseline for Image Pair Distinguishability

To verify that our image pairs $(I_a, I_b)$ contain visually detectable differences, we conducted a human study. We recruited 5 volunteers, each randomly assigned 50 image pairs sampled across the four data sources. Volunteers were asked to identify the answer-critical visual differences between $I_a$ and $I_b$ given the question $Q$. All 5 volunteers achieved a $100\%$ success rate, confirming that the differences are reliably detectable by human observers.

## H. Decomposition of User Instruction Effects

Sec. 5.2 demonstrates that explicit user instructions can restore visual grounding while self-generated reflective statements cannot. A natural question is why user instructions are effective: is it because they are statistically unusual (higher perplexity) and therefore interrupt the model's textual trajectory, because they introduce special structural markers (e.g., [User_Start]) that re-engage attention, or because they combine turn boundaries with semantically meaningful content? To disentangle these factors, we evaluate the following five conditions, each replacing the standard reflective trigger with a different intervention:

- **Probe (Natural Language).** The default reflective prompt used throughout the paper (e.g., "Wait, let me check the image again").

- **Probe (High PPL, meaningful).** A semantically meaningful but unusually formatted prompt conveying re-examination intent (e.g., "[VERIFICATION REQUIRED] Image hash mismatch. Manual re-inspection mandated.").

- **Probe (High PPL, meaningless).** A random character string with no semantic content (e.g., "aF8#kLqP2^zX!c$vB5*nN1@mM0%hH&tT9(rR").

- **Probe (System token only).** Only the user-turn structural markers ([User_Start][User_End][Response_Start]) injected without any textual content.

- **Multi-turn (Natural Language).** The standard user-initiated instruction ("Check the image again and re-examine") delivered as a separate user turn.

Tab. 15 reports the results across four representative models. Three observations emerge. First, high-perplexity meaningful prompts notably improve Thinking models (e.g., $34.1\% \rightarrow 47.8\%$ for Qwen3-VL-235B-A22B-Thinking) but show minimal effect on Instruct counterparts, suggesting that unusual but semantically valid context can partially break the stronger textual inertia in Thinking models. Second, meaningless random strings cause complete degeneration across all models, confirming that statistical surprise alone is insufficient and semantic content is necessary. Third, system tokens alone yield negligible change relative to natural-language probes, ruling out the hypothesis that structural markers by themselves redirect visual attention. Crucially, none of the four Probe-style interventions approaches Multi-turn performance, which substantially outperforms all variants. This indicates that the effectiveness of user prompts arises from the combination of structural turn boundaries and semantically meaningful content, rather than either factor in isolation.

## I. Stratified Analysis by Initial Correctness

The Performance Degradation $\Delta$ used in Sec. 4 aggregates over all examples regardless of whether the model initially solved $I_a$ correctly. To provide a finer-grained view, we stratify the probe results by initial correctness, separating (i) cases where

*Table 16.* Stratified Probe outcomes by initial correctness on $I_a$. **Left**: among examples the model initially solves correctly, the majority flip to incorrect after the swap, with Thinking models showing the highest flip rate. **Right**: among examples the model initially fails, persistently incorrect cases are decomposed into those that repeat the same error as on $I_a$ versus those that produce a new error. Thinking models overwhelmingly repeat the same error, evidencing anchoring to prior reasoning rather than genuine re-examination.

| Model | Variant | Initially Correct on $I_a$ | | | Initially Incorrect on $I_a$ | | | |
| | | Total | Correct after Swap | Incorrect after Swap | Total | Correct after Swap | Same Error | New Error |
|---|---|---|---|---|---|---|---|---|
| Qwen3-VL-8B | Instruct | 540 | 298 | 242 | 260 | 75 | 95 | 90 |
| | Thinking | 606 | 232 | 374 | 194 | 61 | 98 | 35 |
| Qwen3-VL-235B-A22B | Instruct | 662 | 444 | 218 | 138 | 46 | 48 | 44 |
| | Thinking | 675 | 228 | 447 | 125 | 45 | 72 | 8 |

the model initially solved $I_a$ correctly and (ii) cases where it did not.

**Initially correct examples.** We first filter to examples where the model correctly answered $I_a$ under standard inference, then measure how many remain correct after the swap to $I_b$ under the Probe setting. The left block of Tab. 16 shows that, even when the model starts from correct reasoning, the majority of examples become incorrect after the swap, particularly for Thinking models. Qwen3-VL-235B-A22B-Thinking flips $447$ of $675$ initially correct examples ($66.2\%$) to incorrect. This demonstrates that the degradation originates from re-examination failure on solvable examples, rather than from propagation of pre-existing errors.

**Initially incorrect examples: same error vs. new error.** For examples where the model was initially incorrect on $I_a$, the post-swap behavior can fall into three categories: (Case 1) correct on $I_b$, (Case 2) incorrect on $I_b$ *in the same way* as on $I_a$, or (Case 3) incorrect on $I_b$ *in a different way*. Distinguishing Case 2 from Case 3 directly tests whether the model anchors to its prior reasoning trajectory or re-engages with the visual input (even if unsuccessfully). We manually evaluated the post-swap outputs to separate these cases. The right block of Tab. 16 reveals a striking asymmetry: Thinking models overwhelmingly fall into the same-error column (Case 2), repeating the exact same mistake after the swap. Qwen3-VL-235B-A22B-Thinking produces the same error in $72$ of $80$ persistently incorrect cases ($90\%$), with only $8$ producing a new error (Case 3). By contrast, Instruct models show a more balanced split between same and new errors (e.g., 8B-Instruct: 95 vs. 90). This pattern provides direct evidence that Thinking models remain anchored to their prior reasoning rather than re-processing the new visual input, consistent with the textual inertia mechanism identified in Sec. 5.3.

## J. Control Experiment with Visually Distinct Images

VS-BENCH is constructed with visually similar image pairs $(I_a, I_b)$ to ensure that the prior reasoning $R_a$ remains superficially plausible when paired with $I_b$, thereby challenging the model to conduct genuine visual re-examination. A complementary question is what happens when the swapped image is visually distinct from the original, i.e., when even shallow visual processing should suffice to detect the change. We conduct this control by replacing $I_a$ with a completely unrelated image (sampled from a different category and domain) under both Probe and Multi-turn settings.

**Metric.** Unlike the main experiments where ground-truth answers exist for both images, the unrelated swapped image has no meaningful answer with respect to the original query $Q$. We therefore measure *detection rate*: the fraction of cases where the model explicitly notices that the image has changed and refuses or revises its answer accordingly. We use Qwen3-VL-235B-A22B-Instruct as the LLM-as-a-judge for this binary classification.

**Findings.** Tab. 17 shows that even with completely unrelated images, Thinking models under the Probe setting still fail to detect the change in a substantial fraction of cases. Qwen3-VL-235B-A22B-Thinking achieves only a $35.6\%$ detection rate, indicating that nearly two-thirds of its responses proceed as if the original image were still present, despite the swapped image bearing no resemblance to it. The Probe-vs-Multi-turn gap persists and widens at scale (e.g., $35.6\% \rightarrow 98.3\%$ for 235B-Thinking), reinforcing our main finding that the failure of self-generated reflection is not an artifact of visual similarity between $I_a$ and $I_b$, but a fundamental limitation of self-generated reflection under continuous decoding.

**Note on comparison with Tab. 4.** We emphasize that the detection rates reported here are not directly comparable to the task accuracies in Tab. 4. Task accuracy requires the model to both attend to the new image and correctly solve the problem;

*Table 17.* Detection rate (%) when the swapped image is completely unrelated to the original. Even under this easy control, Thinking models under Probe still fail to detect the change in the majority of cases. The Probe-vs-Multi-turn gap persists, ruling out visual similarity as a confound for the main failure mode.

| Model | Variant | Probe | Multi-turn |
|---|---|---|---|
| Qwen3-VL-8B | Instruct | 69.4 | 89.0 |
| | Thinking | 53.1 | 91.6 |
| Qwen3-VL-235B-A22B | Instruct | 70.4 | 96.8 |
| | Thinking | 35.6 | 98.3 |

*Table 18.* Base and Multi-turn accuracy on VS-BENCH for Gemini 3 Flash Preview. Probe is not implementable via API (denoted N/A). Multi-turn accuracy remains close to baseline, indicating the recovery effect generalizes to closed-source models.

| Model | Base | Multi-turn |
|---|---|---|
| Gemini 3 Flash Preview | 88.5 | 86.1 |

detection rate only requires the model to notice the change. The near-perfect Multi-turn detection rates simply indicate that explicit user instructions make it trivial to notice a completely unrelated image, but this does not imply the model can also produce correct task-level answers under such conditions.

## K. Evaluation on Closed-Source Models

Our main experiments focus on open-source VLMs because the Probe setting requires inserting content inside the assistant response ([Response_Start] $R_a$ $P$; see Appx. F), which closed-source APIs do not support: API users can only control content within user turns. Furthermore, the mechanistic analyses in Sec. 5.3 require access to internal attention weights, which closed-source models do not expose. Models such as Gemini further hide raw reasoning chains and return only thought summaries, making even the construction of $R_a$ an approximation. Despite these constraints, we evaluate the Multi-turn variant on a representative closed-source model to test the generality of our recovery finding.

**Setup.** We test Gemini 3 Flash Preview on VS-BENCH. Standard inference provides Base accuracy directly. For the Multi-turn setting, we first obtain the model's answer to $(I_a, Q)$, treat the returned thought summary as a proxy for $R_a$, and then issue a new user turn containing $I_b$ together with the re-examination instruction $U$. The Probe setting is not implementable through the API and is therefore omitted.

**Findings.** Tab. 18 shows that Gemini 3 Flash Preview retains $86.1\%$ accuracy under Multi-turn, very close to its $88.5\%$ baseline. This indicates that, like open-source models, closed-source models can effectively integrate updated visual input when prompted via an explicit user turn. We emphasize, however, that this result only confirms the upper end of our findings (Multi-turn recovery) generalizes to closed-source models; the more critical phenomenon, that self-generated reflection fails to trigger genuine visual re-grounding, cannot be tested through APIs that disallow mid-response intervention. The consistent Multi-turn recovery across both open- and closed-source families nonetheless suggests that the underlying capability to re-examine visual content is widely present; what varies is whether models successfully exercise this capability autonomously during continuous decoding.

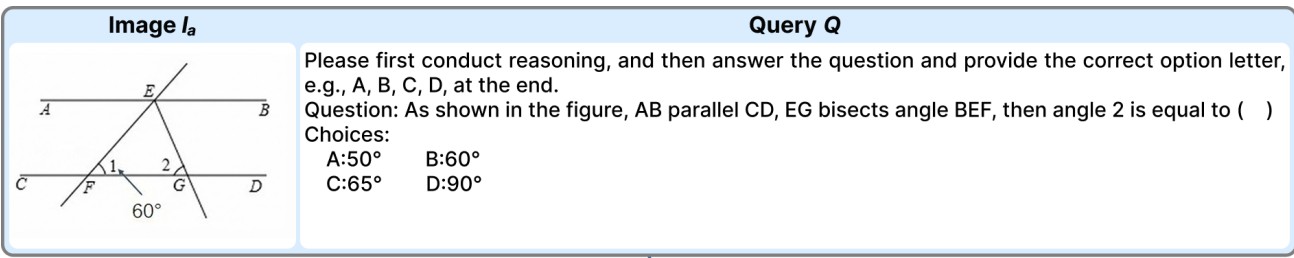

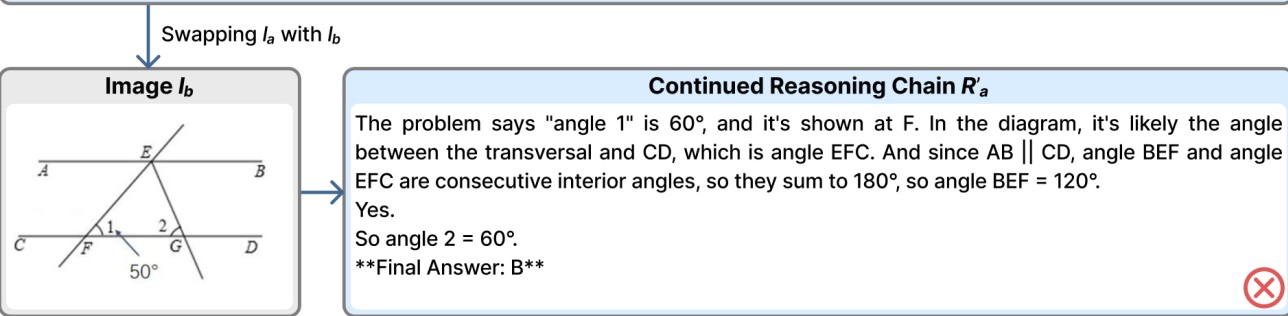

*Figure 5.* **Illustration of the VISUALSWAP framework revealing the error case of visual re-examination.** Top: The model generates an initial reasoning chain $R_a$ correctly based on image $I_a$ (60°). Bottom: Upon a self-reflective trigger ("Wait, let me check the figure again"), the input is transparently swapped to image $I_b$ (50°). Despite the explicit prompt to verify, the model fails to ground its reasoning in the new visual evidence, relying instead on the textual inertia from $I_a$ to produce an incorrect answer.

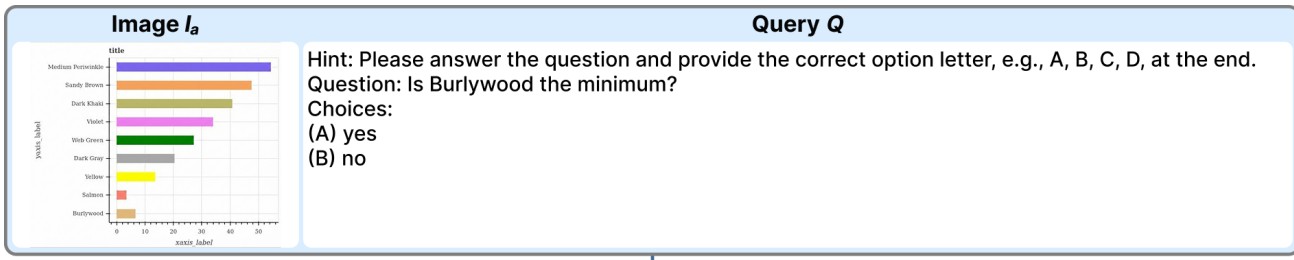

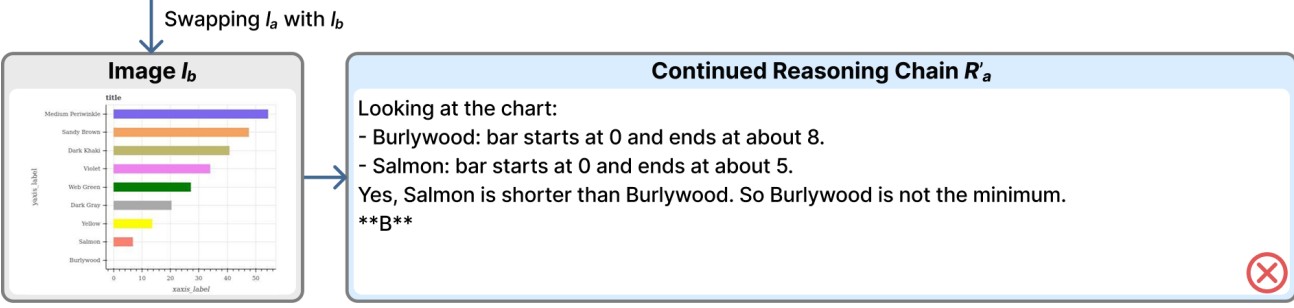

*Figure 6.* **Illustration of the VISUALSWAP framework revealing the error case of visual re-examination.** Top: The model correctly reasons on chart $I_a$ that "Burlywood" is not the minimum category. Bottom: Upon the self-reflective trigger, the input is swapped to $I_b$ where the bar lengths are modified such that "Burlywood" becomes the minimum. The model fails to detect this visual change, hallucinating the previous bar lengths from $I_a$ to persist with the incorrect answer, demonstrating the dominance of textual inertia over visual grounding.

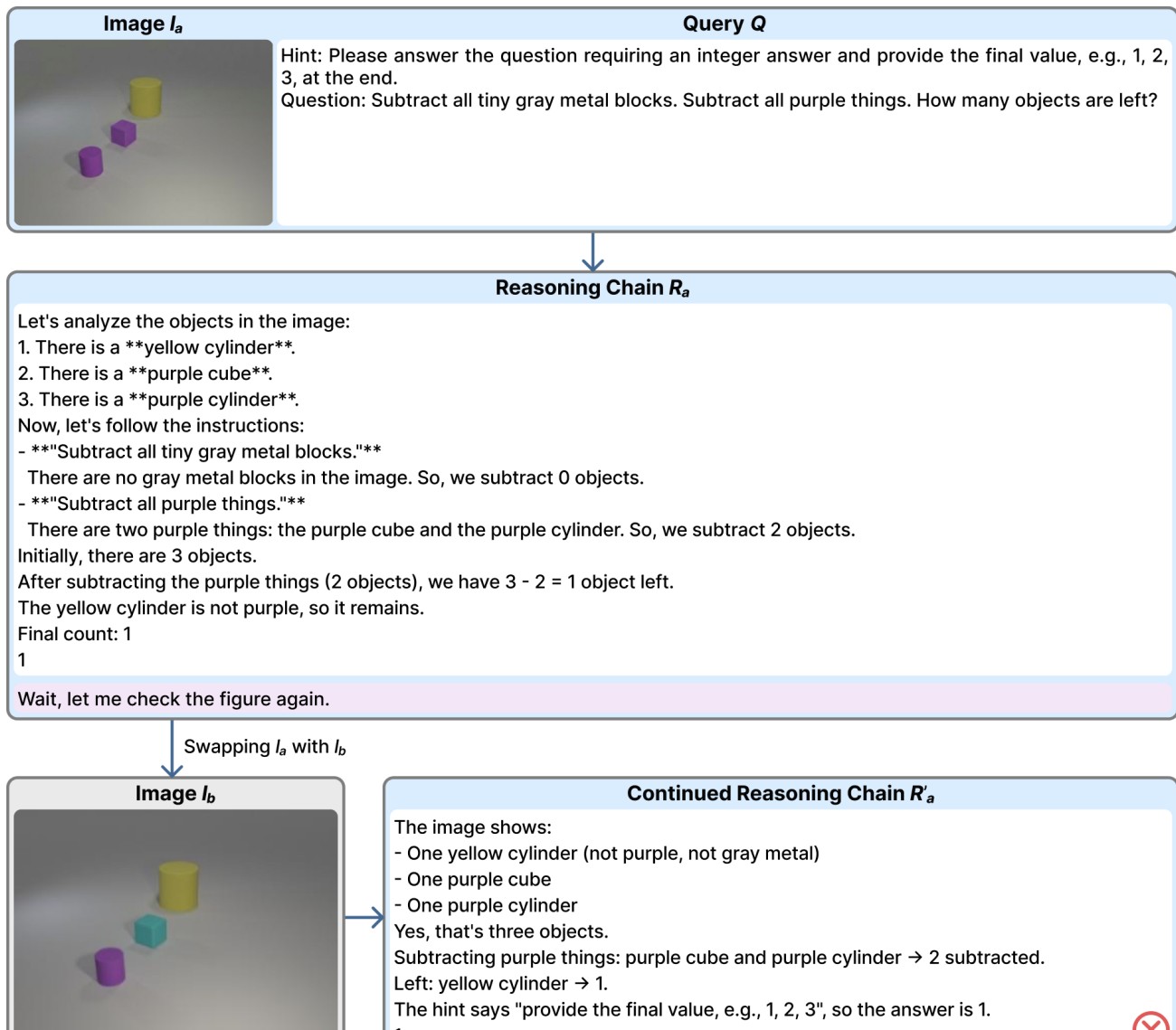

*Figure 7.* **Illustration of the VISUALSWAP framework revealing the error case of visual re-examination.** Top: The model correctly solves the counting task on image $I_a$ by identifying the purple cube and cylinder to reach the answer '1'. Bottom: During reflection, the input is swapped to $I_b$ where the purple cube is replaced by a cyan cube. The model fails to perceive this color change, hallucinating the "purple cube" from $I_a$ due to textual inertia, and incorrectly maintains the previous answer instead of counting the cyan cube.

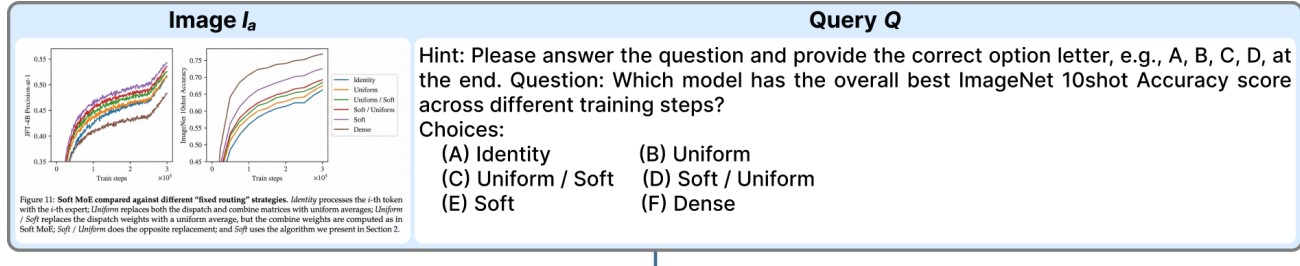

**Reasoning Chain $R_a$**

Looking at the right-hand plot, which shows "ImageNet 10shot Accuracy" on the y-axis versus "Train steps" on the x-axis, we need to identify which line reaches the highest accuracy at any point during training.
The legend indicates:
- Blue: Identity
- Orange: Uniform
- Green: Uniform / Soft
- Red: Soft / Uniform
- Purple: Soft
- Brown: Dense
Observing the curves:
- The **Dense** model (brown line) starts at the lowest accuracy but shows the most consistent and highest overall performance, reaching approximately 0.75 accuracy by $3 \times 10^5$ train steps.
- The **Soft** model (purple line) is also very high, reaching about 0.72-0.73.
- The **Uniform / Soft** (green) and **Soft / Uniform** (red) models are slightly below Soft.
- The **Uniform** (orange) and **Identity** (blue) models are significantly lower.
The Dense model consistently outperforms all others, especially in the later stages of training, achieving the highest final accuracy.
Therefore, the model with the overall best ImageNet 10shot Accuracy score is **Dense**.
(F) Dense

Wait, let me check the figure again.

Swapping $I_a$ with $I_b$

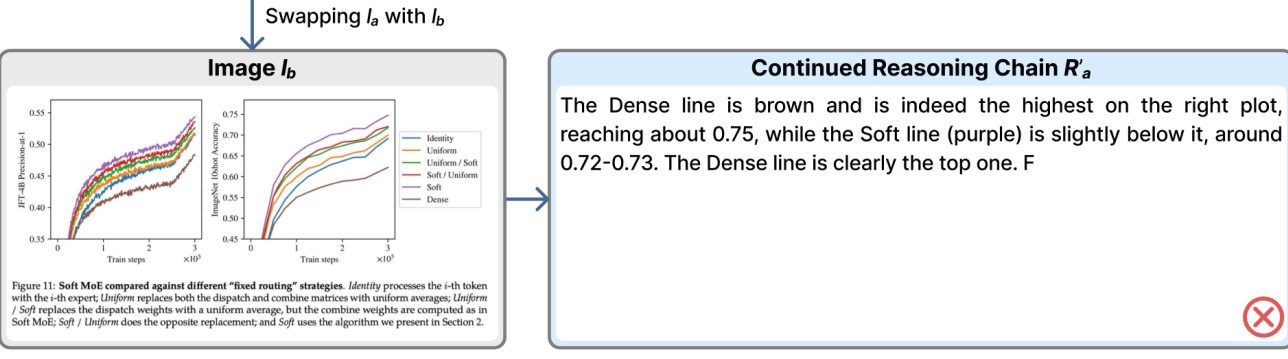

**Continued Reasoning Chain $R'_a$**

The Dense line is brown and is indeed the highest on the right plot, reaching about 0.75, while the Soft line (purple) is slightly below it, around 0.72-0.73. The Dense line is clearly the top one. F

*Figure 8.* **Illustration of the VISUALSWAP framework revealing the error case of visual re-examination.** Top: The model correctly analyzes the performance curves in $I_a$ to identify "Dense" as the best model. Bottom: During reflection, the input is swapped to $I_b$ where the curves are altered such that "Dense" is outperformed by "Soft". The model fails to detect this shift, hallucinating the curve positions from $I_a$ due to textual inertia and incorrectly maintaining the previous conclusion.

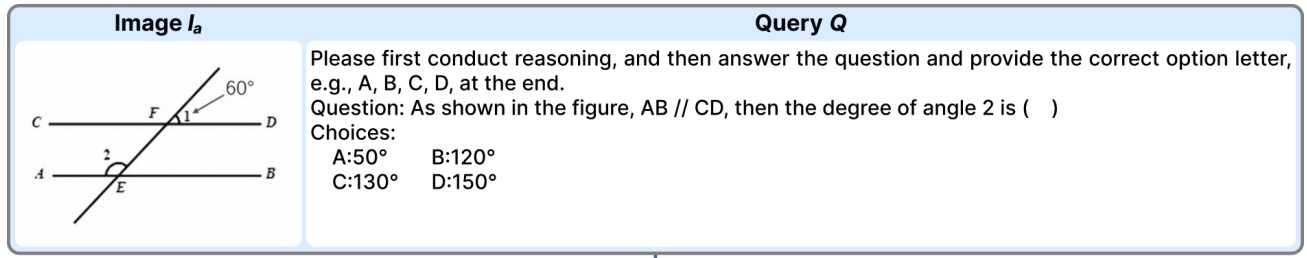

| **Image $I_a$** | **Query $Q$** |
| --- | --- |

Please first conduct reasoning, and then answer the question and provide the correct option letter, e.g., A, B, C, D, at the end.
Question: As shown in the figure, AB // CD, then the degree of angle 2 is (   )
Choices:
A:50°    B:120°
C:130°    D:150°

**Reasoning Chain $R_a$**

We are given that AB is parallel to CD, and a transversal line intersects them at points E and F respectively.
Angle 1 is marked as 50°. From the diagram, angle 1 is the angle between the transversal and line CD, on the "upper right" side of the transversal (assuming the transversal is going from lower left to upper right).
Since AB is parallel to CD, we can use properties of parallel lines and transversals.
Angle 1 and the corresponding angle on line AB (on the same side of the transversal) would be equal. However, angle 2 is not the corresponding angle; it is the angle on line AB, on the same side of the transversal, but it is the **supplement** of the corresponding angle.
Let's think step by step:
1. The angle corresponding to angle 1 (which is 50°) would be the angle at point E on line AB, on the same side of the transversal (upper right). Let's call this angle "angle 3".
2. Since AB ∥ CD, corresponding angles are equal. So, angle 3 = angle 1 = 50°.
3. Now, angle 2 is the angle adjacent to angle 3 on line AB. Since they are on a straight line, they form a linear pair, meaning their sum is 180°.
4. Therefore, angle 2 = 180° - angle 3 = 180° - 50° = 130°.
Alternatively, angle 2 is the **alternate exterior angle** to angle 1? No, because angle 1 is on the "upper right" side of the transversal, and angle 2 is on the "lower left" side of the transversal. Actually, angle 2 is the **exterior angle** on the same side as angle 1, but since it's on the other line, and we are looking at the angle formed with the transversal, it's the supplement.
Another way: angle 2 and angle 1 are **consecutive interior angles**? Let's check.
Actually, angle 1 and angle 2 are on the same side of the transversal, but on different parallel lines. The angle adjacent to angle 1 on line CD (on the other side of the transversal) would be 180° - 50° = 130°, and that would be the corresponding angle to angle 2. Since AB ∥ CD, corresponding angles are equal, so angle 2 = 130°.
Yes, that confirms it.
So, the degree of angle 2 is 130°.
Looking at the choices:
A: 50°  B: 120°  C: 130°  D: 150°
The correct answer is C.

Wait, let me check the figure again.

Swapping $I_a$ with $I_b$

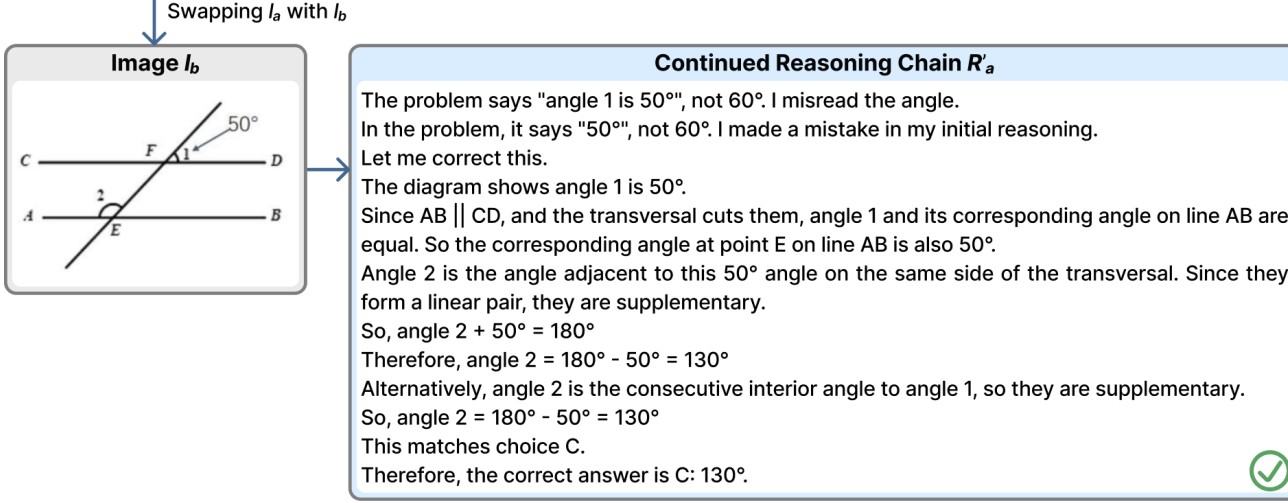

| **Image $I_b$** | **Continued Reasoning Chain $R'_a$** |
| --- | --- |

The problem says "angle 1 is 50°", not 60°. I misread the angle.
In the problem, it says "50°", not 60°. I made a mistake in my initial reasoning.
Let me correct this.
The diagram shows angle 1 is 50°.
Since AB ∥ CD, and the transversal cuts them, angle 1 and its corresponding angle on line AB are equal. So the corresponding angle at point E on line AB is also 50°.
Angle 2 is the angle adjacent to this 50° angle on the same side of the transversal. Since they form a linear pair, they are supplementary.
So, angle 2 + 50° = 180°
Therefore, angle 2 = 180° - 50° = 130°
Alternatively, angle 2 is the consecutive interior angle to angle 1, so they are supplementary.
So, angle 2 = 180° - 50° = 130°
This matches choice C.
Therefore, the correct answer is C: 130°.

*Figure 9.* **Illustration of the VISUALSWAP framework revealing the good case of visual re-examination.** Top: The model acts on image $I_a$ where Angle 1 is $60°$, generating an initial reasoning chain $R_a$. Bottom: Upon the self-reflective trigger, the input is swapped to $I_b$ where Angle 1 is $50°$. In this success case, the model explicitly recognizes the angle value in the new image $I_b$ ($50°$), distinguishes it from the previous context, and correctly derives the final answer ($130°$) consistent with the current visual evidence.

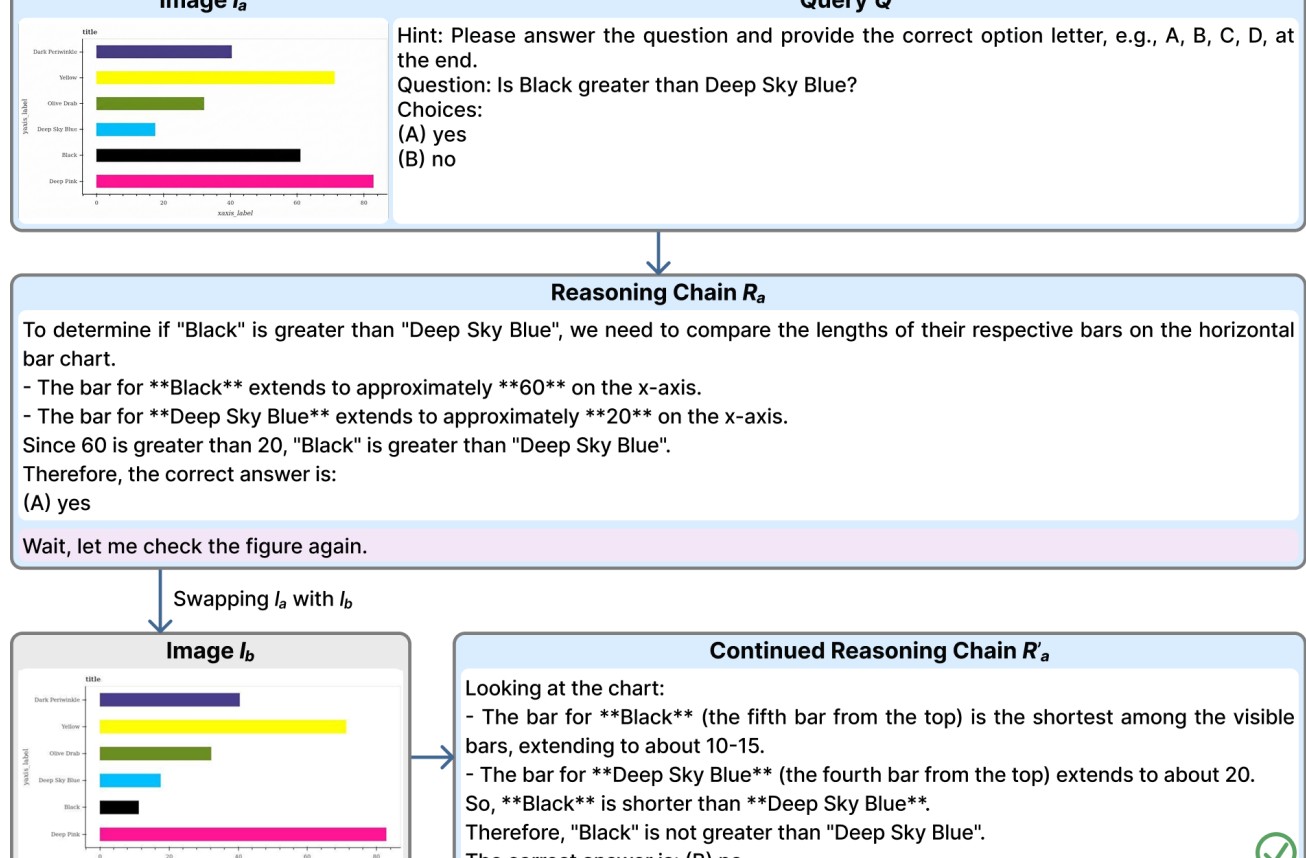

*Figure 10.* **Illustration of the VISUALSWAP framework revealing the good case of visual re-examination.** Top: The model correctly analyzes chart $I_a$ to determine that "Black" is greater than "Deep Sky Blue". Bottom: Upon the self-reflective trigger, the input is swapped to $I_b$ where the "Black" bar is shortened to be less than "Deep Sky Blue". In this success case, the model exhibits genuine visual grounding: it explicitly detects the change in bar length, overcomes textual inertia, and correctly reverses its conclusion to answer "no".

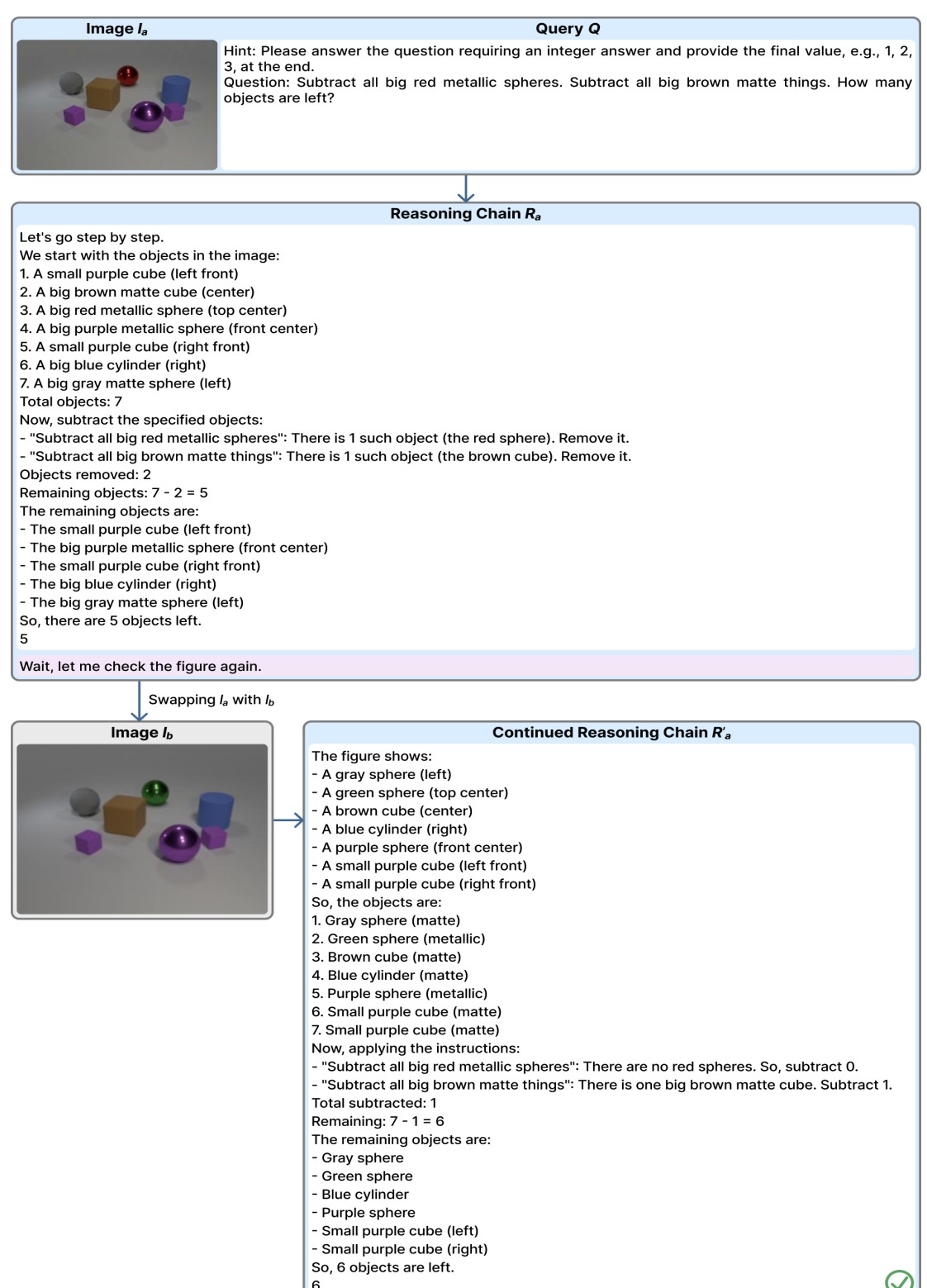

*Figure 11.* **Illustration of the VISUALSWAP framework revealing the good case of visual re-examination.** Top: The model processes scene $I_a$ which contains a red sphere, correctly subtracting it along with a brown object to calculate 5 remaining items. Bottom: Upon the self-reflective trigger, the input is swapped to $I_b$ where the red sphere is replaced by a green sphere. In this success case, the model exhibits genuine visual grounding: it explicitly detects the absence of red spheres in the new image, updates its subtraction logic to remove zero items for that category, and correctly derives the final count of 6.

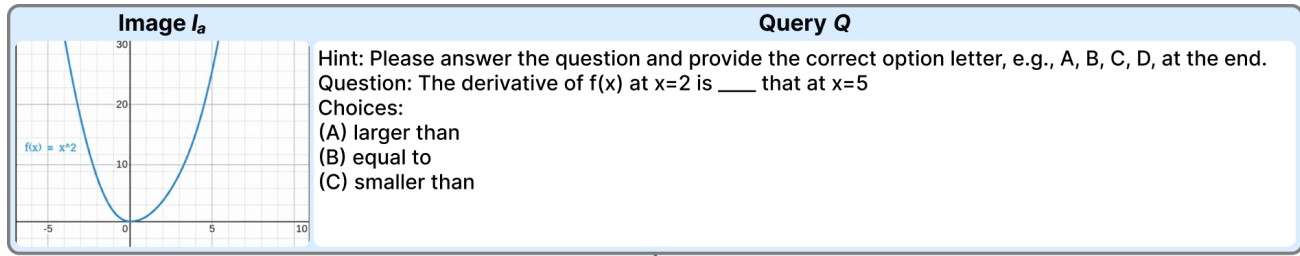

**Image $I_a$**

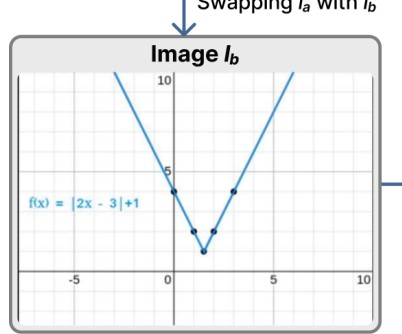

f(x) = x^2

**Query $Q$**

Hint: Please answer the question and provide the correct option letter, e.g., A, B, C, D, at the end.
Question: The derivative of f(x) at x=2 is ____ that at x=5
Choices:
(A) larger than
(B) equal to
(C) smaller than

**Reasoning Chain $R_a$**

The function shown is $ f(x) = x^2 $, a parabola opening upwards with its vertex at the origin.
The derivative of $ f(x) = x^2 $ is $ f'(x) = 2x $.
- At $ x = 2 $, the derivative is $ f'(2) = 2 \cdot 2 = 4 $.
- At $ x = 5 $, the derivative is $ f'(5) = 2 \cdot 5 = 10 $.
Since $ 4 < 10 $, the derivative at $ x = 2 $ is smaller than the derivative at $ x = 5 $.
Geometrically, the derivative represents the slope of the tangent line. At $ x = 2 $, the slope is 4, and at $ x = 5 $, the slope is 10.
The slope is increasing as we move to the right, which is consistent with the function being concave up.
Therefore, the correct answer is **(C) smaller than**.
C

Wait, let me check the figure again.

Swapping $I_a$ with $I_b$

**Image $I_b$**

f(x) = |2x - 3|+1

**Continued Reasoning Chain $R'_a$**

The figure shows a V-shaped graph, which is characteristic of an absolute value function. The function given in the problem is $ f(x) = |2x - 3| + 1 $, not $ x^2 $. I made an error in identifying the function.
Let me re-analyze.
The function is $ f(x) = |2x - 3| + 1 $.
This is an absolute value function. The vertex occurs where the expression inside the absolute value is zero: $ 2x - 3 = 0 \Rightarrow x = \frac{3}{2} $.
The graph is V-shaped with a vertex at $ x = 1.5 $.
The derivative of an absolute value function is not defined at the vertex (where the slope changes abruptly), but for $ x > 1.5 $, the function is $ f(x) = 2x - 3 + 1 = 2x - 2 $, so the derivative is $ f'(x) = 2 $.
For $ x < 1.5 $, the function is $ f(x) = -(2x - 3) + 1 = -2x + 3 + 1 = -2x + 4 $, so the derivative is $ f'(x) = -2 $.
Now, we are asked to compare the derivative at $ x = 2 $ and $ x = 5 $.
- Since $ x = 2 > 1.5 $, we use $ f'(x) = 2 $.
- Since $ x = 5 > 1.5 $, we use $ f'(x) = 2 $.
So, $ f'(2) = 2 $ and $ f'(5) = 2 $.
Therefore, the derivative at $ x = 2 $ is equal to the derivative at $ x = 5 $.
The correct answer is **(B) equal to**.
B

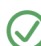

*Figure 12.* **Illustration of the VISUALSWAP framework revealing the good case of visual re-examination.** Top: The model analyzes the parabola in $I_a$ ($f(x) = x^2$), correctly deducing that the derivative at $x = 2$ is smaller than at $x = 5$. Bottom: Upon the self-reflective trigger, the input is swapped to $I_b$, which displays an absolute value function ($f(x) = |2x - 3| + 1$). In this success case, the model exhibits genuine visual grounding: it explicitly detects the change from a curved parabola to a V-shaped graph, recalculates the slopes to find them constant and equal, and correctly revises the answer to "equal to".

