# OpenReview forum: "Are VLMs Seeing or Just Saying? Uncovering the Illusion of Visual Re-examination"
_ICML.cc/2026/Conference — ICML 2026 spotlight_

### Official Review · Reviewer_Zr2p · 2026-03-09

**Soundness:** 3
**Presentation:** 3
**Significance:** 4
**Originality:** 4
**Overall Recommendation:** 5
**Confidence:** 4

**Summary:**

The paper studies whether VLMs genuinely re-examine the visual inputs during self-reflective reasoning. A diagnostic framework called VisualSwap is introduced, it swaps an initial image with an alternative image mid-generation to test if the model notices the change. A dataset VS-BENCH with 800 image pairs is introduced to evaluate multiple VLMs. The results indicate that the models fail to detect the visual swap most of the time, relying on their prior text generation. This is also supported by the attention analysis showing that the models pay little attention to the image during self-reflection.

**Compliance With Llm Reviewing Policy:**

Affirmed.

**Final Justification:**

The added experiments in rebuttal address some of my main concerns. I believe that the paper after revision offers insights that others could build on. I would like to increase my score to "Accept".

**Key Questions For Authors:**

- As mentioned above, how often does the model self-reflect and how is it handled when it happens?
- The degradation metric is defined as the difference between base accuracy and probe accuracy. There's no mention of filtering the dataset to exclusively evaluate instances where the model correctly solved the initial image. If the model is incorrect for the initial image, it's possible for the model to be 1) correct for the swapped image, 2) incorrect for the swapped image, in the same way as initial image; 3) incorrect for the swapped image, but in a different way compared to the initial image. How is this impacting the aggregated results?

**Limitations:**

See weaknesses.

**Strengths And Weaknesses:**

Strengths:
- The paper move beyond behavioral benchmarks by including an insightful attention analysis. By demonstrating that attention to visual tokens remains flat during the reflection, the core argument of the paper is strengthened.
- The discovery that thinking models exhibit much greater vulnerability to reasoning inertia than standard instruct models is a highly valuable insight for the community.
- The paper proves that explicit multi-turn user interventions successfully break this inertia, providing actionable mitigations.

Weaknesses:
- The potential issue with forced injection of reflection. The paper claim to test self-reflective behaviors, but the method forces the reflection. The string "Wait, let me check..." is explicitly appended to the generated reasoning. While 10 variants are tested to ensure robustness against specific phrasing, it does not address the underlying question. Did the authors observe the model do self-reflection during experiments? If so, do we still append the string? If not, isn't forcing a reflection make the behavior out-of-distribution thus unnaturally for the model? A more rigorous approach would half generation only when the model naturally outputs a reflective token.
- Vague/Ambiguous implementation details. There's no details offered in the paper on exactly how the "replace original image with alternative image" is implemented. Assuming using vLLM framework, it's unclear if a custom kernel is utilized to hot-swap visual embeddings in the active KV cache, or if a hard re-prefill of the entire context window with the new image and old text. The distinction is whether the model is computing the new image from scratch but still ignoring it in favor of the text.
- The benchmark uses visually similar image pairs so that the prior reasoning remains superficially plausible. However, by doing so, the authors miss a crucial control experiment where the swapped image is visually distinct. This would give us insight if textual inertia completely overrides all visual perception, or it's only when the model (incorrectly) believes the visual has not changed.

---

> ### Author Rebuttal · Authors · 2026-03-29
>
> We thank the reviewer for the constructive feedback. We address each question below.
> > W1 & Q1: Forced injection of reflection and natural self-reflection frequency
>
> We verified that self-reflection is prevalent across both model types. For Thinking models, we directly counted the occurrence of reflective triggers (e.g., "wait") in their natural reasoning chains. For Instruct models, since they do not undergo long-CoT training, we explicitly instructed them via the system prompt to reflect after producing an answer, then counted the same triggers. We verified that adding this system prompt affects Instruct models' baseline accuracy by less than 2%, confirming it does not alter their reasoning performance. The reflection frequencies are:
>
> |  | 8B-Inst | 8B-Think | 235B-Inst | 235B-Think |
> | --- | --- | --- | --- | --- |
> | Reflection frequency | 87.1 | 93.0 | 90.9 | 96.9 |
>
> This confirms that our probing setup is in-distribution.
>
> Second, we conducted a follow-up experiment where, for models that naturally produced a reflective statement (i.e., generating "wait"), we swapped the image at that natural reflection point; for models that did not produce a natural reflection, we fell back to the standard probing injection. The results are:
>
> |  | 8B-Inst | 8B-Think | 235B-Inst | 235B-Think |
> | --- | --- | --- | --- | --- |
> | Standard probing | 46.6 | 36.6 | 61.3 | 34.1 |
> | Natural reflection | 34.9 | 35.2 | 53.1 | 35.3 |
>
> The natural reflection setting yields even lower accuracy than our standard probing, further strengthening our conclusions. This also rules out the concern that forced injection introduces artificially high perplexity that confuses the model. In fact, as we showed in our response to Reviewer ynGf (Q1), injecting high-perplexity meaningful prompts actually improves performance. The reason we adopted the standardized probing approach is that the linguistic formats of self-reflection are too diverse to capture exhaustively, and controlled injection enables fair cross-model comparison.
>
> > W2 & Q1: Implementation details of image swap
>
> We perform a full re-prefill of the entire context with the new image $I_b$ and the previously generated text tokens. This means the model recomputes fresh visual representations from $I_b$ from scratch, and has full access to the new visual information when continuing generation. We will detail this in the revised manuscript and release the full codebase upon acceptance.
>
> > Q2: Control experiment with visually distinct images
>
> We thank the reviewer for this excellent suggestion. We replaced the original image with a completely unrelated image and used LLM-as-a-judge to evaluate whether the model detects the change:
>
> |  | Probe | Multi-turn |
> | --- | --- | --- |
> | 8B-Inst | 69.4 | 89.0 |
> | 8B-Think | 53.1 | 91.6 |
> | 235B-Inst | 70.4 | 96.8 |
> | 235B-Think | 35.6 | 98.3 |
>
> Even with completely unrelated images, Thinking models under Probe still fail to detect the change in a substantial proportion of cases (e.g., only 35.6% for 235B-Think). The Probe vs. Multi-turn gap persists, reinforcing that the core phenomenon: models "saying" without "seeing" is not an artifact of visual similarity between image pairs, but a fundamental limitation of self-generated reflection.
>
> > Q3: Impact of initially correct and incorrect answers on aggregated results
>
> We thank the reviewer for this question. The concern is whether the degradation metric Δ conflates genuine re-examination failure with propagation of initial errors. To disentangle these factors, we conducted stratified analyses across 4 models. We first filtered to examples where the model correctly solved the original image Ia under standard inference, then measured how many become incorrect after swapping to Ib under the probe condition:
>
> | Model | Initially correct on $I_a$ | Correct after swap | Incorrect after swap |
> | --- | --- | --- | --- |
> | 8B-Instruct | 540 | 298 | 242  |
> | 8B-Think | 606 | 232 | 374  |
> | 235B-Instruct | 662 | 444 | 218 |
> | 235B-Think | 675 | 228 | 447  |
>
> Even when models start from correct reasoning, the majority of examples become incorrect after the swap, particularly for Thinking models. This confirms that the degradation is not driven by propagation of initial errors.
>
> For completeness, we also report results for initially incorrect examples:
>
> | Model | Initially incorrect on $I_a$ | Correct after swap | Incorrect after swap |
> | --- | --- | --- | --- |
> | 8B-Instruct | 260 | 75 | 185 |
> | 8B-Thinking | 194 | 61 | 133 |
> | 235B-Instruct | 138 | 46 | 92 |
> | 235B-Thinking | 125 | 45 | 80 |
>
> Notably, the stratified results also reinforce our main finding that Thinking models are more vulnerable than Instruct counterparts. We will include this analysis in the revised manuscript.
>
> We again thank the reviewer for the constructive suggestions.

---

> > ### Author Rebuttal · Reviewer_Zr2p · 2026-04-03
> >
> > "Q2: control experiment with visually distinct images" (this is for W3, not Q2), the added experiment here deserves a much closer looking into, because I believe it reveals some new insights.
> >
> > Compare the results to Table 4,
> > - All probe results are quite a bit higher.
> > - All multi-turn results are much higher, almost reaching 100.
> > - All multi-turn results are higher than Probe and Base, which is different than Table 4 trend.
> > - The Probe result for 8B instruct exceeds the Base result.
> >
> > All of these indicate that the models behave much differently for visually distinct images and for similar images. This should brought new insights and explanations to the mechanism of what's observed in the paper.
> >
> > Minor point: for Q2 the added table is very helpful, but does not exactly address the questions (especially case 2 and 3).

---

> > > ### Author Response · Authors · 2026-04-04
> > >
> > > We thank the reviewer for the thoughtful follow-up. The observation about the visually distinct control experiment is very insightful, and we appreciate the opportunity to clarify and provide deeper analysis.
> > >
> > > > Control experiment with visually distinct images
> > >
> > > We would like to clarify an important distinction: the visually distinct experiment and Table 4 measure different things and are not directly comparable. In Table 4, we report **task accuracy**, which requires the model to both attend to the new image and correctly solve the problem. In the visually distinct experiment, the swapped image is completely unrelated to the original query, so there is no meaningful ground-truth answer. Instead, we only measure **whether the model notices the image has changed**.
> > >
> > > This difference naturally explains the patterns the reviewer observed. For example, a model may successfully detect that the image has changed, but still fail to correctly solve the new problem. Only when the model can both detect the change and solve the new task would the two metrics be equal. Therefore, detection rates being higher than task accuracies in Table 4 is expected. The near-perfect multi-turn detection rates simply indicate that explicit user instructions make it trivial to notice a completely unrelated image, but this does not imply the model can also solve the new task correctly.
> > >
> > > That said, this control experiment strengthens our main conclusion. Even when the swapped image is completely unrelated, Thinking models under the Probe setting still fail to detect the change in a substantial proportion of cases (e.g., only 35.6% detection rate for Qwen3-VL-235B-A22B-Thinking), and the Probe vs. Multi-turn gap persists across all models. This confirms that models "saying" without "seeing" is not an artifact of visual similarity between image pairs but a fundamental limitation of self-generated reflection.
> > >
> > > > Case 2 and Case 3 among initially incorrect examples
> > >
> > > We thank the reviewer for pointing this out. We conducted additional manual evaluation to distinguish between:
> > >
> > > - **Case 2**: incorrect after swap in the same way as the initial image (same error)
> > > - **Case 3**: incorrect after swap in a different way (new error)
> > >
> > > | Model | Initially incorrect | Correct after swap | Case 2 (same error) | Case 3 (new error) |
> > > | --- | --- | --- | --- | --- |
> > > | Qwen3-VL-8B-Instruct | 260 | 75 | 95 | 90 |
> > > | Qwen3-VL-8B-Thinking | 194 | 61 | 98 | 35 |
> > > | Qwen3-VL-235B-A22B-Instruct | 138 | 46 | 48 | 44 |
> > > | Qwen3-VL-235B-A22B-Thinking | 125 | 45 | 72 | 8 |
> > >
> > > A clear pattern emerges: Thinking models overwhelmingly fall into Case 2 (same error), especially Qwen3-VL-235B-A22B-Thinking (72 vs. 8). This means that when Thinking models are initially wrong, they tend to repeat the exact same mistake after the image swap, confirming that they remain anchored to their prior reasoning rather than processing the new visual input. By contrast, Instruct models show a more balanced split between Case 2 and Case 3, consistent with their relatively weaker textual inertia. This further supports our main finding that Thinking models are more severely affected by reasoning inertia, and that in both cases, models are not genuinely re-examining the visual input. This is consistent with our main results (Sec. 4.2, Tab. 2) showing that Thinking models are less likely to genuinely re-examine the visual input than Instruct counterparts.
> > >
> > > We hope these analyses have addressed the reviewer's remaining concerns.

---

### Official Review · Reviewer_hp2a · 2026-03-09

**Soundness:** 2
**Presentation:** 3
**Significance:** 3
**Originality:** 3
**Overall Recommendation:** 4
**Confidence:** 3

**Summary:**

This paper studies whether VLMs truly re-attend to visual inputs during self-reflective reasoning. The authors propose VISUALSWAP, which swaps the image with a visually similar but semantically different one during reasoning, and introduce VS-BENCH, a benchmark of 800 image pairs with answer-critical differences. Experiments on 15 VLMs show that many models fail to notice the change and continue their original reasoning, suggesting that apparent visual re-examination often results from textual reasoning inertia rather than genuine visual grounding.

**Compliance With Llm Reviewing Policy:**

Affirmed.

**Final Justification:**

This paper presents a novel and insightful investigation into whether VLMs genuinely perform visual re-examination during self-reflective reasoning. The proposed VISUALSWAP framework is simple yet effective, and the empirical results consistently reveal a meaningful failure mode where models rely on textual reasoning inertia rather than true visual grounding. The analysis further provides useful insights into the underlying mechanisms.

There are still some limitations, including the lack of evaluation on closed-source models, potential mismatch between the probing setup and natural reasoning behavior, and the exploration of mitigation strategies. However, the authors have addressed all concerns in the rebuttal, which improves the soundness and of significance the work. I encourage the authors to further explore and strengthen the mitigation strategies in the revised version.

Overall, the paper identifies an important issue and supports it with solid empirical evidence. I find the contribution meaningful and raise my score to 4 (weak accept).

**Key Questions For Authors:**

1. How often do humans fail to distinguish the image pairs? Is there a human baseline verifying that the differences are reliably detectable?
2. Could the dataset be extended beyond math reasoning to general perception tasks?
3. The paper claims thinking models are more vulnerable. Could this simply be due to longer reasoning contexts increasing textual inertia?

**Limitations:**

The proposed framework relies heavily on standard and existing techniques, offering minimal conceptual innovation over prior work.

**Strengths And Weaknesses:**

### Strengths
- The paper introduces an underexplored question: whether multimodal reasoning models actually re-attend to visual inputs during self-reflection. This is a meaningful issue for interpretability, reliability, and multimodal reasoning.
- The experiments cover multiple model families, different scales, and both Instruct and Thinking variants.
- The attention analysis provides a plausible explanation.

### Weaknesses
- While closed-source models do not allow mechanism-level analysis, the proposed VISUALSWAP protocol and VS-BENCH are behavioral evaluations that could still be applied via API testing. It would help verify whether the observed lack of effective visual re-examination generalizes beyond the open-source models.
- The paper shows that explicit user instructions can restore performance close to baseline. While insightful, this raises the question of whether the issue is primarily a limitation of continuous decoding, rather than multimodal reasoning itself.
- The Re-examination Probe setup appears somewhat different from how VLMs normally perform self-reflective reasoning, which is typically generated as a continuous reasoning process. It is unclear whether the VISUALSWAP intervention aligns with the model’s natural inference dynamics, raising questions about whether the probe accurately reflects how models perform visual re-examination in practice.
- While the paper identifies and analyzes the phenomenon of false visual re-examination in VLMs, it does not propose methods or strategies to mitigate this issue.

---

> ### Author Rebuttal · Authors · 2026-03-29
>
> We sincerely thank the reviewer for the detailed review. We appreciate the recognition of the significance of our research question. We believe all concerns can be satisfactorily addressed, as we elaborate below.
> > W1: Evaluation on closed-source models
>
> We appreciate this suggestion. However, applying VISUALSWAP to closed-source models faces fundamental technical barriers (Sec 4.1 Line182-195). Our framework requires:
>
> (1) access to intermediate reasoning chains for identifying reflection points, yet closed-source models increasingly hide raw thoughts (e.g., Gemini only returns thought summaries, not full reasoning traces [1]);
>
> (2) controlled prompt continuation with swapped images mid-generation, which closed-source APIs do not support;
>
> (3) internal attention weights for mechanistic analysis (Sec. 5.3). These limitations are not specific to our work but reflect a broader challenge for diagnostic VLM research as closed-source models become more opaque.
>
> Despite these barriers, our evaluation already covers 15 models across diverse model families, including Qwen3-VL-235B-A22B, one of the strongest open-source VLMs available. Notably, our finding that scaling provides no mitigation (Sec. 4.2, Lines 253-261) suggests that this phenomenon is unlikely to disappear simply by using larger or more capable closed-source models, further reinforcing the generalizability of our conclusions.
>
> [1] https://ai.google.dev/gemini-api/docs/thinking
> > W2: Is this primarily a limitation of continuous decoding rather than multimodal reasoning?
>
> We thank the reviewer for raising this distinction. We would like to respectfully suggest that precisely identifying continuous decoding as the locus of failure is itself a central contribution of our work, rather than a limitation. This carries significant practical implications, as virtually all current reasoning-enhanced VLMs rely on continuous decoding for their reasoning capabilities. Our finding that self-reflective statements within this paradigm fail to trigger genuine visual re-grounding, while semantically equivalent user instructions succeed, reveals a previously unidentified structural asymmetry in how VLMs process self-generated versus externally provided text. Our attention analysis (Tab. 5, Lines 318-332) provides direct mechanistic evidence: user instructions elicit more than double the visual attention compared to self-generated reflections (e.g., Δ=2.21 vs. Δ=1.07 at Layer 55) under the same context. We believe this insight is actionable and directly informs the design of future reasoning VLMs.
>
> > W3: The probe setup differs from natural self-reflective reasoning
>
> Our standardized probing is designed for fair cross-model comparison. We have also conducted experiments under natural self-reflection conditions, and the results are consistent with our reported findings. Due to the character limit, please refer to our response to Reviewer Zr2p (W1) for full details.
>
> > W4: Discussion on mitigation strategies.
>
> We will add a dedicated mitigation discussion in the revised manuscript. Due to the character limit, please refer to our response to Reviewer ynGf (W1 and Q3) for the full discussion.
>
> > Q1: Human baseline for image pair distinguishability
>
> Our human-in-the-loop annotation pipeline (Sec. 3.3, Lines 175-188) inherently ensures detectability: annotators iteratively refined each image pair until the answer-critical differences were clearly identifiable. To further validate this, we recruited 5 volunteers, each randomly assigned 50 image pairs, and all achieved 100% success rate in identifying the differences. We also provide 8 illustrative examples in Appendix A (Figs. 5-12) for inspection.
>
> > Q2: Extension beyond math reasoning
>
> We would like to clarify that VS-Bench already includes general perception tasks. As detailed in Sec. 3.2 (Lines 170-174), MMMU-Pro covers expert-level multimodal understanding across diverse professional disciplines beyond mathematics (examples in Figs. 7 and 11). The consistent failure patterns across all task types suggest that the phenomenon is not specific to math reasoning.
>
> > Q3: Are thinking models more vulnerable simply due to longer context?
>
> We thank the reviewer for this question. Longer context is indeed a contributing factor, and we explicitly analyze this in Sec. 5.4 (Lines 333-344, Fig. 4). However, our results suggest it is not the sole explanation. First, at the same context retention ratio, thinking models exhibit consistently steeper accuracy decline than instruct counterparts (Fig. 4). Second, multi-turn recovers Qwen3-VL-235B-A22B-Thinking from 34.1% to 85.4% (Tab. 4) with full context preserved. If context length were the primary cause, this recovery should not occur. This confirms that the vulnerability stems from how self-generated reasoning suppresses visual attention, not merely from its length.
>
> We again thank the reviewer for the feedback.

---

> > ### Author Rebuttal · Reviewer_hp2a · 2026-04-02
> >
> > (1) For the evaluation on closed-source models:
> > - While we acknowledge the limitations in accessing internal reasoning and attention, our concern is primarily about behavioral generalization. A simplified multi-turn variant of VISUALSWAP, which does not rely on intermediate reasoning traces or mid-generation intervention, could still be applied via APIs. It would be valuable to clarify whether such evaluation was attempted or considered.
> > - Furthermore, it is unclear how the image swapping is implemented (e.g., hidden-state intervention vs. multi-turn interaction), and how reflection points are identified in your experiments, especially for Instruct models that do not produce explicit reasoning traces. Clarifying these details would help assess whether the proposed probing protocol can be adapted to API-based settings, and thus whether evaluation on closed-source models is feasible.
> >
> > (2) For the mitigation strategies:
> > - While the paper provides insightful analysis, the proposed mitigation strategies in the rebuttal remain high-level and are not empirically validated. It would be helpful to know whether the authors have considered implementing  these strategies to validate their effectiveness.

---

> > > ### Author Response · Authors · 2026-04-04
> > >
> > > We thank the reviewer for the follow-up questions. We are glad that our rebuttal has resolved the majority of the concerns (5 out of 7), and we are happy to address the remaining points below.
> > >
> > > > Regarding implementation details and evaluation on closed-source models:
> > >
> > > **How image swapping works.** All experiments use a straightforward re-prefill protocol with no hidden-state intervention. The three settings are:
> > >
> > > **Standard Inference:**
> > >
> > > - Input: `[User_Start][$I_a$][Q][User_End]`
> > > - Output: `[Response_Start][$R_a$][Response_End]`
> > >
> > > **Re-examination Probe** ($R_a$ and $P$ are within the same assistant response):
> > >
> > > - Input: `[User_Start][$I_b$][Q][User_End][Response_Start][$R_a$][P]`
> > > - Output: `[$R_b$][Response_End]`
> > >
> > > **Multi-turn** (Ra is closed as a complete response, U is a new user turn):
> > >
> > > - Input: `[User_Start][$I_b$][Q][User_End][Response_Start][$R_a$][Response_End][User_Start][U][User_End]`
> > > - Output: `[Response_Start][$R_b$][Response_End]`
> > >
> > > In both Probe and Multi-turn, we perform a full re-prefill with the new image $I_b$, meaning the model recomputes fresh visual representations from scratch. The process is illustrated in Fig. 1, with additional examples in Figs. 5-12.
> > >
> > > **How reflection points are identified.** The treatment is identical for Instruct and Thinking models: we append the Re-examination Probe directly after the model's response. This is in-distribution behavior — we verified that models naturally produce reflective triggers in 87-97% of cases, and swapping images at natural reflection points yields consistent results. We adopted standardized appending for fair cross-model comparison. Due to word count limitations, please refer to our response to Reviewer Zr2p (W1 & Q1) for full details.
> > >
> > > **Why closed-source evaluation is infeasible.** As shown above, the Probe setting requires inserting content inside `[Response_Start]...[Response_End]`, which closed-source APIs do not support — we can only control content within `[User_Start]...[User_End]`. Furthermore, models like Gemini only expose thought summaries rather than full $R_a$, making even Multi-turn an approximation. Nonetheless, we tested Gemini 3 Flash Preview:
> > >
> > > |  | Base | Probe | Multi-turn |
> > > | --- | --- | --- | --- |
> > > | Gemini 3 Flash Preview | 88.5 | N/A | 86.1 |
> > >
> > > The multi-turn recovery pattern generalizes to closed-source models. However, this only confirms that closed-source models benefit from Multi-turn behavior. The more critical discovery, that self-generated reflection fails to trigger genuine visual re-grounding, cannot be tested via API.
> > >
> > > > Regarding empirical validation of mitigation strategies:
> > >
> > > We respectfully note that our paper already provides one empirically validated mitigation: the multi-turn intervention (Sec. 5.2, Tab. 4), which recovers Qwen3-VL-235B-A22B-Thinking from 34.1% to 85.4%, nearly matching baseline, and can be automated as a system-level flow without actual user involvement.
> > >
> > > To directly address the reviewer's concern that our mitigation strategies "remain high-level and are not empirically validated," we conducted an additional experiment. Under the Probe setting, we explicitly double the attention weights allocated to visual tokens during generation of $R_b$. (This requires disabling flash attention, causing OOM for Qwen3-VL-235B-A22B, so we report Qwen3-VL-8B only.)
> > >
> > > |  | Probe | Double-attention |
> > > | --- | --- | --- |
> > > | Instruct | 46.6 | 54.5 |
> > > | Thinking | 36.6 | 54.8 |
> > >
> > > Simply amplifying visual attention at the reflection point substantially improves performance, particularly for the Thinking model (+18.2%). This provides direct empirical evidence that insufficient visual attention, as identified in our attention analysis (Sec. 5.3, Figs. 2-3), is a key contributing factor, and that strengthening it during reflection can effectively mitigate the problem. Scaling this to a full training-time solution would require tens of thousands of annotated examples, which constitutes a complete study on its own. We view this as an important follow-up enabled by our diagnostic findings.
> > >
> > > We hope these responses have addressed the reviewer's remaining concerns.

---

### Official Review · Reviewer_ynGf · 2026-03-12

**Soundness:** 4
**Presentation:** 4
**Significance:** 4
**Originality:** 3
**Overall Recommendation:** 5
**Confidence:** 5

**Summary:**

This paper identifies a fundamental **hallucination issue** in existing methods: the commonly introduced **“self-reflection”** process does not genuinely guide the model to re-perceive visual information, but instead serves as a **formalized textual behavior**.
To investigate this phenomenon, the authors propose a new benchmark, **VS-Bench**, which evaluates whether models truly reconsider the visual input by **replacing the target image during the re-examination stage**. Through experiments, the authors further observe that **user prompts in multi-turn dialogues can effectively guide the model to reconsider its reasoning**, suggesting that **external interventions can significantly influence the model’s behavior**.
Overall, I believe the paper provides **sufficient experimental evidence** and presents a **reasonable motivation**, and therefore my rating is **Accept**.

**Compliance With Llm Reviewing Policy:**

Affirmed.

**Final Justification:**

The authors actively discuss with me and address all of my concerns. I keep my rating (*accept*)

**Key Questions For Authors:**

See the content in **Weakness** and **Discussion**

**Limitations:**

Yes

**Strengths And Weaknesses:**

**Strength**
1. The paper reveals an important phenomenon: models often **pretend to perform reflection** rather than genuinely re-examining the visual content.
2. The authors find that **User Prompts can significantly guide the model’s attention**, enabling it to reprocess visual information more effectively. This observation is particularly interesting and highlights an aspect that has largely been overlooked in previous evaluations. It also suggests that the **purity and structure of the context** may play an important role in influencing model behavior.
3. The authors introduce a new benchmark, **VS-Bench**, to evaluate whether models can truly re-examine visual information. This benchmark could provide useful insights for the community in studying reflection behavior in multimodal models.
4. The experimental evaluation is **comprehensive and convincing**.

---

**Weakness**
1. Although the paper exposes the hallucinated “re-examination” phenomenon, it lacks a deeper discussion on **how future work could address or mitigate this issue**, or how model architectures and training strategies might be improved to encourage genuine visual reconsideration.

---

**Discussion**
1. Since **User Prompts significantly influence model behavior**, it is worth investigating the underlying reasons. I personally see two possible perspectives:
    - Considering that the User Prompt is provided as an **instruction input**, it may act as an **unusual context** whose joint probability with the existing context is low. From this perspective, does this suggest that introducing **rare or unexpected context information** may help redirect the model’s attention?

    - In some models (e.g., **Qwen2.5-VL**), user prompts are marked with special tokens such as `"<im_start>user"`. Could such **consistent structural markers** play a role in explicitly guiding the model’s attention?

2.  Do the authors plan to **open-source VS-Bench**?
3.  If explicit **external interventions** can significantly improve accuracy, what potential approaches could incorporate such interventions into **SFT or RL training pipelines**?

---

> ### Author Rebuttal · Authors · 2026-03-29
>
> We sincerely thank the reviewer for the positive and insightful review. We are encouraged by the recognition of the importance of our research question, the comprehensiveness of our evaluation, and the value of VS-Bench for the community. We address each question below.
> > W1: Discussion on mitigation strategies.
>
> R1: While our current manuscript primarily emphasizes diagnosis over mitigation, we believe that identifying the root cause is an essential prerequisite for effective solutions. We agree with the need for actionable takeaways and will incorporate a new section detailing these mitigation strategies in the final version.
>
> Based on our findings, we propose the following concrete directions:
>
> 1. Our discovery that multi-turn interaction successfully restore visual grounding while self-generated reflections cannot (Sec. 5.2-5.3, Tab. 4-5) suggests an inference-time strategy: when models need to re-examine visual content, the system can simulate a user-initiated re-examination turn rather than relying on the model's self-generated reflection within continuous decoding. This effectively converts the failing self-reflection paradigm into the succeeding multi-turn paradigm without actual user involvement.
> 2. The success of multi-turn interaction (Sec. 5.2, Tab. 4) motivates a training-time approach: incorporating multi-turn visual verification data into SFT/RL pipelines, where models are explicitly trained to detect discrepancies between prior reasoning and updated visual input.
> 3. Our attention analysis (Sec. 5.3, Figs. 2-3) reveals that self-generated reflective statements fail to elicit sufficient visual token activation. This suggests incorporating visual attention strength as an auxiliary reward signal during RL training, encouraging models to genuinely re-attend to visual tokens when generating reflective statements, rather than merely producing the linguistic form of re-examination.
>
> > Q1: Why do user prompts work — unusual context vs. special tokens?
>
> We compared 5 conditions:
>
> (1) Probe: Natural Language: the standard reflective prompt used in our paper (e.g., "Wait, let me check the image again");
>
> (2) Probe: High PPL (meaningful): a semantically meaningful but unusually formatted prompt conveying re-examination intent (e.g., "[VERIFICATION REQUIRED] Image hash mismatch. Manual re-inspection mandated.");
>
> (3) Probe: High PPL (meaningless): a random character string with no semantic content (e.g., "aF8#kLqP2^zX!c$vB5*nN1@mM0%hH&tT9(rR");
>
> (4) Probe: System token only: injecting only the user-turn structural markers (`<|im_start|>user<|im_end|><|im_start|>assistant`) without any textual content;
>
> (5) Multi-turn: Natural Language: the user-initiated re-examination instruction used in our paper (i.e., "Check the image again and re-examine"), delivered as a separate conversation turn.
>
> |  | Qwen3-VL-8B-Instruct | Qwen3-VL-8B-Thinking | Qwen3-VL-235B-A22B-Instruct | Qwen3-VL-235B-A22B-Thinking |
> | --- | --- | --- | --- | --- |
> | Probe: Natural Language | 46.6 | 36.6 | 61.3 | 34.1 |
> | Probe: High PPL (meaningful) | 48.4 | 43.3 | 63.1 | 47.8 |
> | Probe: High PPL (meaningless) | 0.0 | 0.0 | 0.0 | 0.0 |
> | Probe: System token only | 48.6 | 33.3 | 64.0 | 37.9 |
> | Multi-turn: Natural Language | 58.2 | 67.5 | 77.9 | 85.4 |
>
>
> Several interesting findings emerge.
>
> 1. High-perplexity meaningful prompts notably improve Thinking models (e.g., 34.1%→47.8% for Qwen3-VL-235B-A22B-Thinking) but show minimal effect on Instruct models, suggesting that unusual but semantically valid context can partially break the stronger textual inertia in Thinking models.
>
> 2. Meaningless random strings cause complete degeneration across all models, confirming that semantic content is necessary and mere statistical surprise is insufficient.
>
> 3. System tokens alone yield negligible change, suggesting that structural markers by themselves cannot redirect visual attention without accompanying meaningful content.
>
> 4. Multi-turn interaction substantially outperforms all probe conditions, indicating that the full combination of structural turn boundaries and semantically meaningful user instructions is the key driver. Neither factor alone fully explains the effectiveness of user prompts. We will include this analysis in the revised manuscript.
>
> > Q2: Open-sourcing VS-Bench?
>
> Yes. We will release the full benchmark, evaluation code, and all related resources upon acceptance.
>
> > Q3: Incorporating interventions into SFT/RL?
>
> As discussed in W1, multi-turn visual verification data can be directly incorporated into training pipelines. VS-Bench itself can serve as a foundation for such data construction. Additionally, visual attention strength can be used as an auxiliary RL reward signal to encourage genuine re-grounding.
>
>
> We again thank the reviewer for the valuable feedback.

---

> > ### Author Rebuttal · Reviewer_ynGf · 2026-04-02
> >
> > Most of my concerns have been addressed, and I currently have no other questions.
> > I keep the rating.

---

> > > ### Author Response · Authors · 2026-04-06
> > >
> > > Thank you very much for your valuable suggestions and for your final acknowledgment. We sincerely appreciate your positive feedback and are glad that our rebuttal was able to address your concerns. We are truly grateful for your support and recognition.

---

### Decision · Program_Chairs · 2026-04-30

**Decision:**

Accept (spotlight)

**Comment:**

This paper presents a compelling investigation into the authenticity of self-reflection in Vision-Language Models (VLMs), introducing the VisualSwap framework and the VS-Bench benchmark to test whether models genuinely re-examine images when they claim to do so. The study uncovers a significant "hallucination" phenomenon termed textual inertia, where models, particularly high-capacity "Thinking" models, tend to rely on their initial textual reasoning rather than updating their understanding based on new visual input. This finding is supported by rigorous attention analysis, which reveals that self-generated reflection tokens often fail to trigger a meaningful increase in visual attention, effectively proving that these models are often "saying" they are looking without actually "seeing."

The submission is a very strong paper, as it addresses a fundamental and timely flaw in the current multimodal reasoning paradigm. The authors successfully navigated the rebuttal phase by providing empirical evidence for mitigation strategies, such as doubling attention weights and utilizing multi-turn interactions to restore visual grounding. With unanimous support from reviewers, the paper is recognized for its technical soundness, impactful benchmarking, and deep mechanistic insights. Its ability to challenge the perceived reliability of long Chain-of-Thought (CoT) reasoning in VLMs makes it a high-priority contribution for the conference.